# WEIGHTED POINT SET EMBEDDING FOR MULTIMODAL CONTRASTIVE LEARNING TOWARD OPTIMAL SIMILARITY METRIC

**Toshimitsu Uesaka[1], Taiji Suzuki[2,3], Yuhta Takida[1], Chieh-Hsin Lai[1], Naoki Murata[1], Yuki Mitsufuji[1,4]**

[1]Sony AI, [2]The University of Tokyo, [3]RIKEN AIP, [4]Sony Group Corporation

`toshimitsu.uesaka@sony.com, taiji@mist.i.u-tokyo.ac.jp,`
`{yuta.takida,chieh-hsin.lai,naoki.murata,yuki.mitsufuji}@sony.com`

## ABSTRACT

In typical multimodal contrastive learning, such as CLIP, encoders produce one point in the latent representation space for each input. However, one-point representation has difficulty in capturing the relationship and the similarity structure of a huge amount of instances in the real world. For richer classes of the similarity, we propose the use of weighted point sets, namely, sets of pairs of weight and vector, as representations of instances. In this work, we theoretically show the benefit of our proposed method through a new understanding of the contrastive loss of CLIP, which we call symmetric InfoNCE. We clarify that the optimal similarity that minimizes symmetric InfoNCE is the pointwise mutual information, and show an upper bound of excess risk on downstream classification tasks of representations that achieve the optimal similarity. In addition, we show that our proposed similarity based on weighted point sets consistently achieves the optimal similarity. To verify the effectiveness of our proposed method, we demonstrate pretraining of text-image representation models and classification tasks on common benchmarks.

## 1 INTRODUCTION

CLIP (Radford et al., 2021) and ALIGN (Jia et al., 2021) established one of the most common frameworks for multimodal representation learning (Guo et al., 2019). In this framework, to obtain the text-image representation, two encoders that map inputs from different modalities onto a shared space are trained with a contrastive loss (Chopra et al., 2005). Recent studies have shown that a CLIP model pretrained on a large-scale text-image dataset provides transferable features to various downstream tasks such as linear classification (Radford et al., 2021; Jia et al., 2021), text-to-video retrieval (Lin et al., 2022), and text-conditioned image generation (Ramesh et al., 2022). Other work has shown that a CLIP model can be used to feed vision information to large language models (Alayrac et al., 2022). In addition to text and image modalities, this multimodal contrastive learning framework can be applied to other combinations of modalities such as text-audio representations (Elizalde et al., 2023) and combinations of more than two modalities (Guzhov et al., 2022; Wu et al., 2022; Girdhar et al., 2023).

Despite the success of CLIP models, it is still arguable whether the similarity structure and representations they provide are suitable for modeling concepts in the real world. Typical CLIP encoders transform each input image or text into one point embedding in a latent space, and encoders are trained to enhance the similarity of paired concepts in a training dataset, which is defined by the cosine similarity of their embeddings. However, concepts in the real world have a broadness that raises the relationship of inclusion and many-to-many correspondences. For example, the text "a photo of dogs" can conceivably be the caption of any number of different images, while another text, "a photo of poodles", could be the caption of the subset of dog photos, and the photo of poodles should be linked to the multiple captions. Considering these relationships, representations of concepts should be provided in a manner that goes beyond a singular point and exhibit innate broadness.

In this paper, we propose the use of a weighted point set, namely a set of pairs of a scalar weight and a vector point, as the representation of each concept, which we call Weighted Point Set Embedding (WPSE). We define the similarity of two weighted point sets with a kernel function that defines the similarity of two points. We also provide a theoretical rationale of the proposed weighted point set embedding through a new understanding of the contrastive loss utilized in CLIP, which we call the symmetric InfoNCE loss. First, we highlight the fact that minimization of the symmetric InfoNCE loss is achieved when the similarity of two features in the loss is represented by the pointwise mutual information. Second, we show, under some assumptions, that the optimal (possibly nonlinear) classifier in downstream classification tasks can be constructed by a linear classifier over learned representations when the optimal similarity is achieved. Last, we show that the proposed similarity of weighted point sets has richer representation capacity than the cosine similarity, which is the bilinear similarity in the latent space. Moreover, to demonstrate the effectiveness of the proposed method, we conduct experiments on the Conceptual Caption datasets and common benchmark datasets.

## 2 RELATED WORK

### 2.1 MULTIMODAL CONTRASTIVE REPRESENTATION LEARNING IN PRACTICE

CLIP (Radford et al., 2021) and ALIGN (Jia et al., 2021) utilize contrastive loss to obtain text-image representations, inspired by a series of studies of deep metric learning and unimodal contrastive learning such as multi-class N-pair loss (Sohn, 2016), InfoNCE (Oord et al., 2018), SimCLR (Chen et al., 2020), and ConVIRT (Zhang et al., 2022). Both works have shown the success of pretrained models with large-scale paired datasets and the contrastive loss, which we call the symmetric InfoNCE in this paper, in zero-shot settings and downstream tasks.

One approach to extending this contrastive framework is to modify the similarity in the symmetric InfoNCE loss. Fürst et al. (2022) proposed using modern Hopfield networks for computing similarities to enrich the covariance structure of data, while also replacing the InfoNCE with the InfoLOOB. Desai et al. (2023) proposed using the Lorentzian distance in a hyperbolic space as the similarity to capture a hierarchy structure of visual and linguistic concepts. Following this approach, we propose enriching the class of the similarity based on a nonlinear kernel and weighted point sets. In contrast to the above studies, we provide an analysis of excess risk in downstream linear classifications.

### 2.2 THEORETICAL UNDERSTANDING OF CONTRASTIVE LOSS

Early works attributed the success of the InfoNCE loss (Oord et al., 2018) to the fact that it is a lower bound of mutual information and its optimization leads to maximization of the mutual information between two views of data (Hjelm et al., 2019; Bachman et al., 2019; Tian et al., 2020). However, Tschannen et al. (2020) demonstrated through a thought experiment and empirical results that maximizing tighter bounds on mutual information can result in worse representations. Li et al. (2021) also showed that different representations with the same mutual information can exhibit different qualities. In an alternative perspective, Wang & Isola (2020) investigated alignment and uniformity to understand the InfoNCE. This idea has affected subsequent works on theoretical analysis of contrastive learning (Li et al., 2021; Zimmermann et al., 2021; Huang et al., 2023).

Regarding the theoretical relationship to downstream tasks, Saunshi et al. (2019) showed that the downstream classification loss is upper bounded by a quantity monotonically increasing with respect to the contrastive loss. Although Saunshi et al. (2019) relied on the strong assumption of the conditional independence of two samples, subsequent studies have mitigated this problem. HaoChen et al. (2021) proposed the spectral contrastive loss and provided an upper bound of the linear probe performance based on the augmentation graph. Tosh et al. (2021) analyzed the excess loss of linear predictors on the landmark embedding from the perspective of multi-view redundancy. Wang et al. (2022) showed upper and lower bounds for downstream performance through the conditional feature variance and the augmentation overlap effect. Ash et al. (2022) investigated upper bounds of a supervised loss in terms of the number of negative samples. Huang et al. (2023) analyzed the performance of the nearest neighbor classifier through $(\sigma, \delta)$-augmentation. Shi et al. (2023) investigated the trade-off between label efficiency and universality under assumptions of linear data. Waida et al. (2023) proposed the kernel contrastive loss and showed an upper bound of the classification error. Chen et al. (2024) studied zero-shot transfer capability of CLIP with an awareness of unexpected positive pairs. Zhai

et al. (2024) analyzed self-supervised representation learning through the lens of RKHS induced by augmentations.

However, we argue that there are still three issues to be resolved in terms of understanding the framework of CLIP. First, some works provided only upper bounds of downstream losses. If there is a certain gap between the upper bounds and the optimal value, reducing the contrastive loss does not always mean a better performance in the downstream task. Second, some works changed the target of theoretical analysis from the actual setting of CLIP or InfoNCE and provided guarantees on their proposed losses or different features from usual contrastive learning. Last, some upper or lower bounds included various statistics (e.g., variance) of the obtained presentations. While such bounds are useful when a perfect alignment is achieved, the perfect alignment is not always practical in the case of multimodal learning, where paired samples are not generated by augmentations of the same instance and a data sample in a modality has relationship to many samples in another modality.

Our work differs from the above studies in the following ways. First, we consider not only an upper bound of the performance but also the gap from the optimal classifier. Second, we analyze the symmetric InfoNCE and linear classifiers that are constructed using an approach similar to the actual setting of CLIP. Last, our assumptions for theoretical results are relatively mild in the case of multimodal representation learning, which is explained in Section 4.2.

## 3 PROBLEM SETUP

In this section, we introduce the notations and problem settings that we use in following sections. We formalize the multimodal contrastive representation learning and the downstream linear classification task , which is commonly utilized to evaluate representation learning methods (Chen et al., 2020; Radford et al., 2021).

### 3.1 CONTRASTIVE REPRESENTATION LEARNING AND SYMMETRIC INFONCE

Let $\mathcal{X}$ and $\mathcal{Y}$ denote the input space of two modalities. For the sake of simplicity, we focus on text-image representation learning, and we denote the image space by $\mathcal{X}$ and the text space by $\mathcal{Y}$. Let $p_{X,Y}(x,y)$ denote the density of the joint data distribution of random variables $(X,Y)$ defined over $\mathcal{X} \times \mathcal{Y}$, and let $p_X(x)$ and $p_Y(y)$ denote the density of the marginal distribution of $X$ and $Y$, respectively. If there is no ambiguity, we omit subscripts of probability (density) functions such as $p(x,y), p(x)$, and $p(y)$. We denote the conditional probability density of $y$ given $x$ as $p_Y(y \mid x)$. For a subset $\mathcal{Y}' \subseteq \mathcal{Y}$, we denote the probability with which $Y \in \mathcal{Y}'$ as $P_Y(\mathcal{Y}') := \int_{y \in \mathcal{Y}'} p(y)\mathrm{d}y$. We also denote the conditional probability of a subset $\mathcal{Y}'$ given $x$ as $P_Y(\mathcal{Y}' \mid x) := \int_{y \in \mathcal{Y}'} p(y \mid x)\mathrm{d}y$. For a probability density function $p$, we denote the support of the probability as $\mathrm{supp}\, p$.

Given a batch of $N$ image-text pairs $(x_1, y_1), \ldots, (x_N, y_N) \sim p_{X,Y}$, CLIP (Radford et al., 2021) introduced the following contrastive loss to train an image encoder $f_{\mathcal{X}} \colon \mathcal{X} \to \mathbb{R}^d$, a text encoder $f_{\mathcal{Y}} \colon \mathcal{Y} \to \mathbb{R}^d$, and a trainable temperature parameter $\tau \in \mathbb{R}_{>0}$.

$$\hat{\mathcal{L}}(f_{\mathcal{X}}, f_{\mathcal{Y}}, \tau) = \frac{1}{2}\left[ -\frac{1}{N}\sum_{i=1}^{N}\ln\frac{\exp\left(f_{\mathcal{X}}(x_i)^{\top}f_{\mathcal{Y}}(y_i)/\tau\right)}{\sum_{k=1}^{N}\exp\left(f_{\mathcal{X}}(x_k)^{\top}f_{\mathcal{Y}}(y_i)/\tau\right)} \right.$$
$$\left. -\frac{1}{N}\sum_{i=1}^{N}\ln\frac{\exp\left(f_{\mathcal{X}}(x_i)^{\top}f_{\mathcal{Y}}(y_i)/\tau\right)}{\sum_{k=1}^{N}\exp\left(f_{\mathcal{X}}(x_i)^{\top}f_{\mathcal{Y}}(y_k)/\tau\right)} \right] \tag{1}$$

We call this the symmetric InfoNCE loss. By minimizing it, the similarity of two features from paired samples $(x_i, y_i)$ is expected to be large, and the similarity of two features from independent samples $x_i$ and $y_j$ $(i \neq j)$ is expected to be small. Here, the similarity of two features is measured by the cosine similarity $f_{\mathcal{X}}(x)^{\top}f_{\mathcal{Y}}(y)$. Note that the features $f_{\mathcal{X}}(x)$ and $f_{\mathcal{Y}}(y)$ of typical CLIP are L2-normalized. For a generalized formulation, we replace the scaled similarity $f_{\mathcal{X}}(x)^{\top}f_{\mathcal{Y}}(y)/\tau$ with a function $g \colon \mathcal{X} \times \mathcal{Y} \to \mathbb{R}$ of two samples $(x,y) \in \mathcal{X} \times \mathcal{Y}$. In addition, following the asymptotic form of the InfoNCE in Wang & Isola (2020), we consider the population expectation form of the symmetric InfoNCE. By considering the limit as $N \to \infty$, we have the population expectation form

of the symmetric InfoNCE:

$$\mathcal{L}_{\mathrm{NCE}}(g) = \frac{1}{2} \mathop{\mathbb{E}}_{p(x,y)} \left[ -\ln \frac{\exp g(x,y)}{\mathbb{E}_{p_X(x')}\left[\exp g(x',y)\right]} \right] + \frac{1}{2} \mathop{\mathbb{E}}_{p(x,y)} \left[ -\ln \frac{\exp g(x,y)}{\mathbb{E}_{p_Y(y')}\left[\exp g(x,y')\right]} \right], \quad (2)$$

where we omit the constant term that comes from $\ln N$.

## 3.2 DOWNSTREAM CLASSIFICATION TASK

As a common evaluation of the learned representations with the symmetric InfoNCE, we consider a supervised classification task with $K$ labels. For an integer $M$, we define $[M] := \{1, \ldots, M\}$. Let $C$ denote a random variable for labels. Let $P_C(c \mid x)$ be the conditional probability of the label $c \in [K]$ given the data $x \in \mathcal{X}$. We define $p(x, c) = P_C(c \mid x)p_X(x)$ as *the density* of the joint distribution of data $x$ and its label $c$. We assume that pairs of data and its label $(x, c)$ can be drawn from $p(x, c)$. In this supervised learning setting, a classifier $h \colon \mathcal{X} \to \mathbb{R}^K$ is often trained by minimization of the softmax cross-entropy loss given by $\mathcal{L}_{\mathrm{sup}}(h) := \mathbb{E}_{p(x,c)} \left[ -\ln \frac{\exp h(x)_c}{\sum_{i=1}^{K} \exp h(x)_i} \right]$, where $h(x)_i$ denotes the $i$-th entry of $h(x) \in \mathbb{R}^K$. In downstream linear classifications after the contrastive learning, $h$ is constructed as a linear classifier over the learned representation. Given the encoder $f_\mathcal{X}$, we formalize this linear classifier as $h(x; f_\mathcal{X}) := W^\top f_\mathcal{X}(x) + b$, where $W \in \mathbb{R}^{d \times K}$ is a weight and $b \in \mathbb{R}^K$ is a bias. With this $h(x; f_\mathcal{X})$, the downstream classification task is formalized as the minimization problem of $\mathcal{L}_{\mathrm{sup}}$ with respect to $W$ and $b$: $\min_{W \in \mathbb{R}^{d \times K}, b \in \mathbb{R}^K} \mathcal{L}_{\mathrm{sup}}(h(x; f_\mathcal{X}))$.

## 4 THEORETICAL GUARANTEE VIA POINTWISE MUTUAL INFORMATION

In this section, we derive the upper bound for the performance of downstream classification tasks. First, we highlight that the optimal similarity of the symmetric InfoNCE loss is represented by the pointwise mutual information. Second, we show that if the optimal similarity is obtained, there is a linear classifier on the learned representation that is close to the optimal (possibly nonlinear) classifier. Last, we investigate an error caused by the deviation from the optimal similarity.

### 4.1 POINTWISE MUTUAL INFORMATION AS OPTIMAL SIMILARITY

Our analysis starts with the following fact that the optimal similarity of the symmetric InfoNCE is represented by the pointwise mutual information (Oord et al., 2018; Zhang et al., 2023).

**Proposition 4.1** (Restatement of Proposition 1 in Zhang et al. (2023))**.** *Let $X$ and $Y$ denote two random variables having the joint probability density $p$. Then, the mutual information of $X$ and $Y$, $I(X, Y) := \mathbb{E}_{p(x,y)} \left[ \ln \frac{p(x,y)}{p(x)p(y)} \right]$ is an upper bound of $-\mathcal{L}_{\mathrm{NCE}}(g)$. Moreover, if the function $g$ satisfies $g(x, y) = \ln \frac{p(x,y)}{p(x)p(y)} + \text{const}$, then the equality $I(X, Y) = -\mathcal{L}_{\mathrm{NCE}}(g)$ holds.*

In other words, when we consider the minimization problem of $\mathcal{L}_{\mathrm{NCE}}(g)$ in terms of the measurable function $g$ over $\mathcal{X} \times \mathcal{Y}$, the optimal similarity is equal to the pointwise mutual information up to a constant. We denote this optimal similarity by $g^*(x, y) := \ln \frac{p(x,y)}{p(x)p(y)} + \Gamma$ for some $\Gamma \in \mathbb{R}$.

### 4.2 POINTWISE MUTUAL INFORMATION ESTIMATOR LEADS TO A GOOD LINEAR CLASSIFIER

Next, we show that, under some conditions, there exists a linear classifier over learned representations that is close to the optimal classifier $h^* = \arg\min_h \mathcal{L}_{\mathrm{sup}}(h)$ if we successfully obtain encoders that achieve the optimal similarity $g^*(x, y)$. It is known that the log probability of the label $c$ conditioned by data $x$ is the minimizer of $\mathcal{L}_{\mathrm{sup}}$ up to a constant: $h^*(x)_i = \ln P_C(i \mid x) + \text{const}$, for $i \in [K]$. This is because $\mathcal{L}_{\mathrm{sup}}$ is represented by using the cross entropy $H(\cdot, \cdot)$ as follows: $\mathcal{L}_{\mathrm{sup}}(h) = \mathbb{E}_{p(x)} \left[ H\big(P_C(C \mid x), Q_C(C \mid x; h)\big) \right]$, where $Q_C(c \mid x; h) := \frac{\exp h(x)_c}{\sum_{i=1}^{K} \exp h(x)_i}$ for $c \in [K]$.

To explain our theoretical results, we define several probability (density) functions. We consider $K$ disjoint subsets $\mathcal{Y}_i$ $(i \in [K]) \subseteq \mathcal{Y}$, i.e., for $i \neq j$, $\mathcal{Y}_i \cap \mathcal{Y}_j = \emptyset$. Let $\tilde{\mathcal{Y}} = \mathcal{Y}_1 \cup \mathcal{Y}_2 \cup \cdots \cup \mathcal{Y}_K$. Note that $\tilde{\mathcal{Y}}$ is not necessarily equal to $\mathcal{Y}$. We assume that $P(\mathcal{Y}_i) \neq 0$ for every $i$. We define the conditional

probability of $y$ given $\mathcal{Y}_i$ as $p_Y(y \mid \mathcal{Y}_i) := \frac{p(y)}{P(\mathcal{Y}_i)}$ if $y \in \mathcal{Y}_i$, otherwise 0. Note that $p_Y(Y \mid \mathcal{Y}_i)$ is a probability density function on $\mathcal{Y}$ (i.e., $\int_{y \in \mathcal{Y}} p_Y(y \mid \mathcal{Y}_i)\mathrm{d}y = 1$). Similarly, we define the conditional probability of $y$ given $x$ and $\mathcal{Y}_i$ as $p_Y(y \mid x, \mathcal{Y}_i) := \frac{p(y|x)}{P(\mathcal{Y}_i|x)}$ if $y \in \mathcal{Y}_i$, otherwise 0. For a label $c \in [K]$, we define the conditional probability of a subset $\mathcal{Y}_c$ given $x$ and the union of disjoint subsets $\tilde{\mathcal{Y}}$ as $P_C(c \mid x; (\mathcal{Y}_i)_{i \in [K]}) := \frac{P_Y(\mathcal{Y}_c|x)}{P_Y(\tilde{\mathcal{Y}}|x)}$. We regard this as a probability function of labels over $[K]$ as $\sum_{c \in [K]} P_C(c \mid x; (\mathcal{Y}_i)_{i \in [K]}) = 1$. Last, we construct a linear classifier on learned representations. Given the disjoint subsets $(\mathcal{Y}_i)_{i \in [K]}$ and the components of similarity $g(x, y) = f_\mathcal{X}(x)^\top f_\mathcal{Y}(y)/\tau$, we define $\bar{h}^g(x) := \bar{W}^\top f_\mathcal{X}(x) + \bar{b}$, with a weight $\bar{W} := [\bar{w}_1, \quad \bar{w}_2, \quad \ldots, \quad \bar{w}_K] \in \bar{\mathbb{R}}^{d \times K}, \bar{w}_i := \mathbb{E}_{p_Y(y|\mathcal{Y}_i)}\left[\frac{1}{\tau} f_\mathcal{Y}(y)\right] \in \mathbb{R}^d$, and a bias $\bar{b} := [\ln P_Y(\mathcal{Y}_1), \quad \ln P_Y(\mathcal{Y}_2), \quad \ldots, \quad \ln P_Y(\mathcal{Y}_K)]^\top \in \mathbb{R}^d$.

Now, we show an upper bound on the excess risk of the downstream classification when we obtain encoders that achieve the optimal similarity of the symmetric InfoNCE.

**Theorem 4.2.** *Let $(\mathcal{Y}_i)_{i \in [K]}$ be any choice of disjoint subsets in $\mathcal{Y}$. Assume that $g^*(x, y) := \frac{1}{\tau^*} f_\mathcal{X}^*(x)^\top f_\mathcal{Y}^*(y) = \ln \frac{p(x,y)}{p(x)p(y)} + \text{const}$ holds for any $x \in \text{supp }p(x) \subseteq \mathcal{X}$ and any $y \in \tilde{\mathcal{Y}}$. Then,*

$$\mathcal{L}_{\text{sup}}(\bar{h}^{g^*}) - \mathcal{L}_{\text{sup}}(h^*) \leq \mathbb{E}_{p(x)}\left[D_{\text{KL}}\left(P_C(C \mid x) \,\middle\|\, P_C(C \mid x; (\mathcal{Y}_i)_{i \in [K]})\right)\right]$$
$$+ \mathbb{E}_{p(x,c)}\left[D_{\text{KL}}(p_Y(Y \mid \mathcal{Y}_c) \,\|\, p_Y(Y \mid x, \mathcal{Y}_c))\right]. \quad (3)$$

We defer the proof to Appendix B.1

**Remark.** The first term in RHS of Eq. (3) becomes zero when, for any $c$ and $x$, the conditional probability $P_Y(\mathcal{Y}_c|x)$ is proportional to the conditional probability of label $P_C(c|x)$. The second term in RHS becomes zero when $y$ is independent of $x$ given a prior knowledge that $y$ is in $\mathcal{Y}_c$. Considering the prompt ensembling in zero-shot classifications (Radford et al., 2021) and the properties of text data, we claim that there exist subsets $(\mathcal{Y}_i)_{i \in [K]}$ that satisfy most of those conditions. To construct a classifier in zero-shot classification, Radford et al. (2021) proposed ensembling embeddings of prompt templates such as "a photo of a {}" and "an example of a {}", where the brackets are replaced with the labels such as "dog" and "cat". Since the set of prompts for each label is generated simply by inserting words representing the label into templates, the probability of each set should be roughly proportional to the probability of the label. In addition, prompt templates lack most of the information specific to images, so each prompt in the set can be considered more or less independent of images. Assuming these properties of the text data domain, the excess risk of the linear classifier is close to zero when trained encoders achieve the optimal similarity, which is the pointwise mutual information.

## 4.3 Excess Risk Analysis via the Gap from the Pointwise Mutual Information

We have observed that a similarity equal to the pointwise mutual information (up to a constant) leads to a small excess risk of linear classifiers on the downstream classification. However, an actual similarity $g(x, y)$ obtained in pretraining is possibly different from $g^*(x, y)$ because of the non-convexity of the optimization problem and the insufficient representational capability of the class of similarity, $\{(x, y) \mapsto f_\mathcal{X}(x)^\top f_\mathcal{Y}(y)/\tau \mid f_\mathcal{X}(x), f_\mathcal{Y}(y) \in \mathbb{R}^d, \tau \in \mathbb{R}_{>0}\}$. To consider the effect of the gap in the similarity, we decompose the risk of the downstream task as follows:

$$\mathcal{L}_{\text{sup}}(\bar{h}^g) - \mathcal{L}_{\text{sup}}(h^*) = \left(\mathcal{L}_{\text{sup}}(\bar{h}^g) - \mathcal{L}_{\text{sup}}(\bar{h}^{g^*})\right) + \left(\mathcal{L}_{\text{sup}}(\bar{h}^{g^*}) - \mathcal{L}_{\text{sup}}(h^*)\right). \quad (4)$$

The second term in RHS of Eq. 4 is already bounded by Theorem 4.2. Regarding the first term, we have the following bound.

**Lemma 4.3.** *Assume that, there exists $\Delta \geq 0$ such that $|g(x, y) - g^*(x, y)| \leq \Delta$ for all $x \in \text{supp }p(x)$ and all $y \in \text{supp }p(y)$. Then, it holds that $\left|\mathcal{L}_{\text{sup}}(\bar{h}^g) - \mathcal{L}_{\text{sup}}(\bar{h}^{g^*})\right| \leq 2\Delta$.*

We defer the proof to Appendix B.2. From Theorem 4.2, Lemma 4.3 and the fact that $\min_{W \in \mathbb{R}^{d \times K}, b \in \mathbb{R}^K} \mathcal{L}_{\text{sup}}(W^\top f_\mathcal{X}(\cdot) + b) \leq \mathcal{L}_{\text{sup}}(\bar{h}^g)$, we have the following result.

**Theorem 4.4.** *Assume that there exist $K$ disjoint subsets $\mathcal{Y}_i$ $(i \in [K]) \subseteq \mathcal{Y}$ such that $D_{\text{KL}}\left(p_C(C \mid x) \,\middle\|\, p_C(C \mid x; (\mathcal{Y}_i)_{i \in [K]})\right) \leq \varepsilon_1$, and $D_{\text{KL}}(p_Y(Y \mid \mathcal{Y}_c) \,\|\, p_Y(Y \mid x, \mathcal{Y}_c)) \leq \varepsilon_2$, for*

*all $x \in \operatorname{supp} p(x)$, for all $c \in [K]$, and for some non-negative constants $\varepsilon_1, \varepsilon_2 \geq 0$. Assume that the uniform approximation error of the optimization problem $\arg\min_g \mathcal{L}_{\mathrm{NCE}}(g)$ is bounded by a constant $\Delta \geq 0$, i.e., $|g(x,y) - g^*(x,y)| \leq \Delta$ for all $x \in \operatorname{supp} p(x)$ and $y \in \operatorname{supp} p(y)$. Then, it holds that*

$$\min_{W \in \mathbb{R}^{d \times K}, b \in \mathbb{R}^K} \mathcal{L}_{\mathrm{sup}}(W^\top f_{\mathcal{X}}(\cdot) + b) - \mathcal{L}_{\mathrm{sup}}(h^*) \leq \varepsilon_1 + \varepsilon_2 + 2\Delta. \tag{5}$$

**Remark.** $\Delta$ indicates the gap between actually obtained similarity $g(x,y)$ and the optimal similarity $g^*(x,y)$, which is the pointwise mutual information. Theorem 4.4 implies that the approximation error of the optimal similarity in pretraining may degrade the performance of downstream classifications.

## 5 AUGMENTED SIMILARITY BY WEIGHTED POINT SETS

We have observed that the optimal similarity of symmetric InfoNCE for pretraining leads to a small excess risk on downstream classifications. Here, a question arises: "To what extent can the class of similarity approximate the pointwise mutual information?" In this section, we show a limitation of the typical similarity that is commonly utilized in CLIP. To overcome the issue, we propose a new class of similarities and show a theoretical guarantee of the approximation capability of the proposed class.

### 5.1 LIMITATION OF THE INNER-PRODUCT SIMILARITY IN FINITE DIMENSIONAL SPACES

Consider a $d$-dimensional feature space. We assume there are $N(> d+1)$ pairs of samples, $(x_1, y_1), \ldots, (x_N, y_N) \in \mathcal{X} \times \mathcal{Y}$. We define $Z_{\mathcal{X}}, Z_{\mathcal{Y}} \in \mathbb{R}^{d \times N}$ as the concatenation of features of samples as $Z_{\mathcal{X}} := [f_{\mathcal{X}}(x_1), \ldots, f_{\mathcal{X}}(x_N)]$ and $Z_{\mathcal{Y}} := [f_{\mathcal{Y}}(y_1), \ldots, f_{\mathcal{Y}}(y_N)]$. During pretraining with the symmetric InfoNCE, the similarity matrix $Z_{\mathcal{X}}^\top Z_{\mathcal{Y}}$ is fit to the optimal similarity matrix $G \in \mathbb{R}^{N \times N}$ up to a constant $\Gamma \in \mathbb{R}$, where $G_{ij} = \ln \frac{p(x_i, y_j)}{p(x_i)p(y_j)}$. Regarding the gap $\Delta$ to the optimal similarity, it holds that $\Delta \geq \sup_{x \in \operatorname{supp} p(x), y \in \operatorname{supp} p(y)} |g(x,y) - g^*(x,y)| \geq \sup_{i,j} |(Z_x^\top Z_y)_{ij} - \Gamma - G_{ij}|$. However, it also holds that $\operatorname{rank}(Z_x^\top Z_y + \Gamma J) \leq d+1$, where $J \in \mathbb{R}^{N \times N}$ is the matrix in which all entries are 1 (See Proposition C.1). Thus, if the rank of $G$ is $N > d+1$, there exists a certain error of the approximation of $G$. In other words, to completely capture the structure of the pointwise mutual information, the dimension of feature $d$ is required to be more than the number of intrinsic instances in the data space, which is infeasible in real-world scenarios.

### 5.2 AUGMENTED SIMILARITY BY A NONLINEAR KERNEL AND WEIGHTED POINT SETS

Increasing the dimension of the feature is the simplest way to enhance the capability of the similarity. However, this often requires a larger deep neural network model, which leads to heavier computation both in the contrastive learning phase and in the downstream tasks. As an alternative approach, we propose enriching the class of similarity by using a nonlinear kernel function and weighted point sets (namely, sets of a pair of weight and point). Figure 1 shows the overview of the proposed method. We replace the similarity in the symmetric InfoNCE with a similarity between two weighted point sets produced by encoders.

Following CLIP, we use two encoders that transform inputs from each modality. Instead of one vector in a latent space, the encoders are modified to produce a weighted point set, namely, a set of $M$ pairs of weight and vector: $\{(w_i, v_i)\}_{i \in [M]}$, where $w_i \in \mathbb{R}$ and $v_i \in \mathbb{R}^d$ for each $i \in [M]$. We define the similarity of two weighted point sets, $\left\{(w_i^{(\mathcal{X})}, v_i^{(\mathcal{X})})\right\}_{i \in [M^{(\mathcal{X})}]}$ and $\left\{(w_i^{(\mathcal{Y})}, v_i^{(\mathcal{Y})})\right\}_{i \in [M^{(\mathcal{Y})}]}$ (containing $M^{(\mathcal{X})}$ and $M^{(\mathcal{Y})}$ pairs of weight and vector, respectively), with a kernel function $k(\cdot, \cdot) : \mathbb{R}^d \times \mathbb{R}^d \to \mathbb{R}$, as follows:

$$g\left(\left\{\left(w_i^{(\mathcal{X})}, v_i^{(\mathcal{X})}\right)\right\}_{i \in [M^{(\mathcal{X})}]}, \left\{\left(w_j^{(\mathcal{Y})}, v_j^{(\mathcal{Y})}\right)\right\}_{j \in [M^{(\mathcal{Y})}]}\right) := \sum_{i,j} w_i^{(\mathcal{X})} w_j^{(\mathcal{Y})} k(v_i^{(\mathcal{X})}, v_j^{(\mathcal{Y})}). \tag{6}$$

This similarity can be regarded as the inner product of high-dimensional representers of a linear combination of Dirac measures (Muandet et al., 2017) as $\sum_{i,j} w_i^{(\mathcal{X})} w_j^{(\mathcal{Y})} k(v_i^{(\mathcal{X})}, v_j^{(\mathcal{Y})}) =$

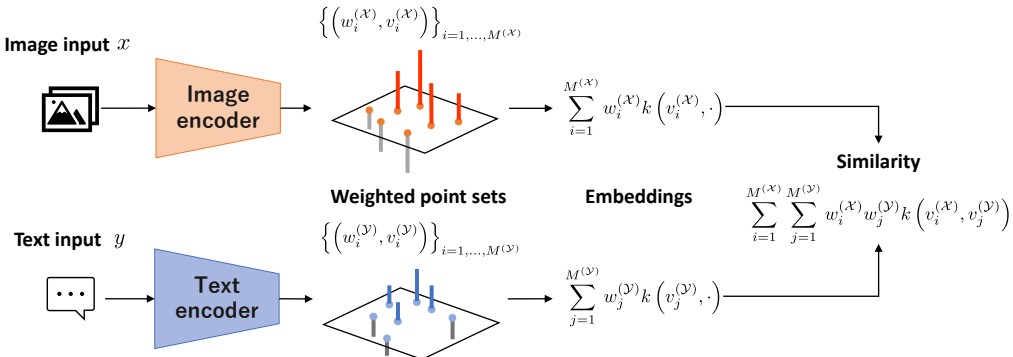

Figure 1: Overview of proposed method. Each encoder produces a weighted point set from each input. The encoders are optimized with the symmetric InfoNCE using the similarity matrix.

$\left\langle \sum_i w_i^{(\mathcal{X})} k(v_i^{(\mathcal{X})}, \cdot), \sum_j w_j^{(\mathcal{Y})} k(v_j^{(\mathcal{Y})}, \cdot) \right\rangle_{\mathcal{H}}$. Here, $\langle \cdot, \cdot \rangle_{\mathcal{H}}$ denotes the inner product of the reproducing kernel Hilbert space (RKHS) associated with $k$. In the following theorem, we show that our proposed similarity based on weighted point sets consistently achieves the optimal similarity.

**Theorem 5.1.** *Assume that Assumption C.2 holds. Define a function $g$ as Eq. 6 with a bounded $c_0$-universal kernel $k \colon \mathbb{R}^d \times \mathbb{R}^d \to \mathbb{R}$. Then, for any $\varepsilon > 0$, there exist positive integers, $M^{(\mathcal{X})}, M^{(\mathcal{Y})} \in \mathbb{N}$ and maps, $f_{\mathcal{X}} : x \mapsto \left\{ \left( w_i^{(\mathcal{X})}, v_i^{(\mathcal{X})} \right) \right\}_{i \in [M^{(\mathcal{X})}]}$ and $f_{\mathcal{Y}} : y \mapsto \left\{ \left( w_j^{(\mathcal{Y})}, v_j^{(\mathcal{Y})} \right) \right\}_{j \in [M^{(\mathcal{Y})}]}$ such that*

$$\sup_{x \in \mathrm{supp}\, p(x), y \in \mathrm{supp}\, p(y)} \left| g(f_{\mathcal{X}}(x), f_{\mathcal{Y}}(y)) - \ln \frac{p(x,y)}{p(x)p(y)} \right| < \varepsilon. \tag{7}$$

The proof and Assumption C.2 are provided in Section C.2. The definition of $c_0$-universal kernel is deferred to Definition C.5 (refer to Sriperumbudur et al. (2011)). For example, the Gaussian kernel $k(u, v) = \exp\left( -\frac{1}{2\sigma^2} \|u - v\|_2^2 \right)$ and the inverse multiquadric (IMQ) kernel $k(u, v) = \frac{c}{\sqrt{c^2 + \|u - v\|_2^2}}$ are $c_0$-universal (Sriperumbudur et al., 2011).

**Remark.** Theorem 5.1 ensures that the proposed class of similarity is capable of approximating the pointwise mutual information in arbitrary precision. Unlike the typical class of similarity discussed in Section 5.1, Assumption C.2 does not require the dimension $d$ proportional to the number of intrinsic instances. Instead, it requires $d$ larger than or equal to the intrinsic dimensions of subspaces of $x \in \mathcal{X}$ and $y \in \mathcal{Y}$ that have dependency on each other. However, we claim that this assumption on $d$ is fairly mild because *the manifold hypothesis* (Bengio et al., 2013) is commonly assumed. Although increasing $M^{(\mathcal{X})}$ and $M^{(\mathcal{Y})}$ also leads to heavy computation, at least it provides a different approach to augmenting representation models than just increasing the number of feature dimensions.

## 5.3 IMPLEMENTATION

To produce a weighted point set from each input, we utilize the structure of transformers, without any significant change to the model size or computation time (Fig. 2). We use Vision Transformer (Dosovitskiy et al., 2021) for the image encoder and Transformer (Vaswani et al., 2017) for the text encoder. A typical Vision Transformer takes projected patches of an image and the special token [CLS], applies attention layers and the last projection layer to the token sequence, and outputs the vector at the position of the [CLS] token. To output additional weights, we add a projection layer for weights in parallel with that for vectors. Moreover, to output a sets of weights and vectors, our image encoder outputs all resultant vectors. In the same way, we modify the text encoder to output all resultant weights and vectors instead of just the vector at the position of a special token [EOS]. In addition, we modify the special tokens of the text encoder for padding, [PAD], to be dependent on its relative position to the [EOS] in order to avoid repeating the same tokens. More specifically, separate padding tokens [PAD1], [PAD2], etc., are appended after the [EOS] until the token sequences reach a fixed length. A learnable embedding is independently initialized for each token.

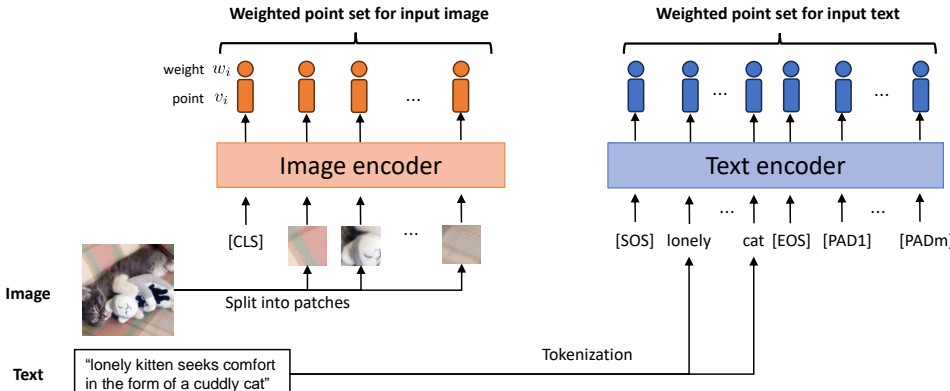

Figure 2: Proposed modification for encoders to produce a weighted point set. Encoders are modeled by Transformer. The encoders output all resultant vectors instead of just one vector at a certain position.

For the kernel function, we opted to use a linear combination of the linear kernel and a nonlinear kernel $\tilde{k}$ with coefficients $\alpha_1, \alpha_2 \in \mathbb{R}_{\geq 0}$: $k(u, v) = \alpha_1 u^\top v + \alpha_2 \tilde{k}(u, v)$. In preliminary experiments, we found that when the model was trained only with a nonlinear kernel (i.e., $(\alpha_1, \alpha_2) = (0, 1)$) the symmetric InfoNCE loss did not decrease well nor converge. We consider this was possibly because of the gradient vanishing for points that were far away from each other. To avoid $O(M^{(\mathcal{X})}M^{(\mathcal{Y})})$ times computation of the kernel, we use random Fourier features (RFFs) (Rahimi & Recht, 2007) for approximating the nonlinear kernel. When the kernel $\tilde{k}$ is shift-invariant, RFF approximates the kernel $\tilde{k}(u, v)$ by the inner product of two $D$-dimensional vectors, i.e. $z(u)^\top z(v) \approx \tilde{k}(u, v)$. $z(v) \in \mathbb{R}^D$ is constructed using random samples $\omega_t \in \mathbb{R}^d$ and $\beta_t \in \mathbb{R}$ $(t = 1, \dots, D)$ from predefined distributions (see Appendix A.1 for details). By taking the weighted sum of RFFs, $\overline{z} := \sum_i w_i z(v_i)$, calculated from points in the point set, we obtain an embedding of the weighted point set. More rigorously, this can be regarded as the embedding in the RKHS of a linear combination of Dirac measures where the RFF approximation is applied to obtain finite dimensional representations of the embeddings. In our implementation, we concatenate the weighted sum of points, $\overline{v} := \sum_i w_i v_i$, and that of RFFs $\overline{z}$, using coefficients $\alpha_1$ and $\alpha_2$ as follows: $\left[\sqrt{\alpha_1}\overline{v}^\top, \sqrt{\alpha_2}\overline{z}^\top\right]^\top$. We use it as an embedding of weighted point sets because we can obtain an unbiased estimator of the similarity in Eq. (6) by simply taking the inner product of embeddings:

$$\sum_{i,j} w_i^{(\mathcal{X})} w_j^{(\mathcal{Y})} k(v_i^{(\mathcal{X})}, v_j^{(\mathcal{Y})}) \approx \sum_{i,j} w_i^{(\mathcal{X})} w_j^{(\mathcal{Y})} \left( \alpha_1 v_i^{(\mathcal{X})\top} v_j^{(\mathcal{Y})} + \alpha_2 z(v_i^{(\mathcal{X})})^\top z(v_j^{(\mathcal{Y})}) \right)$$

$$= \begin{bmatrix} \sqrt{\alpha_1}\overline{v}^{(\mathcal{X})} \\ \sqrt{\alpha_2}\overline{z}^{(\mathcal{X})} \end{bmatrix}^\top \begin{bmatrix} \sqrt{\alpha_1}\overline{v}^{(\mathcal{Y})} \\ \sqrt{\alpha_2}\overline{z}^{(\mathcal{Y})} \end{bmatrix}. \tag{8}$$

It is worth noting that the dimension $D$ of RFFs can be changed between training and inference, which affects the kernel approximation error. For example, a larger $D$ can be used during pretraining to achieve a lower variance in approximation, and a smaller $D$ can be used during inference to reduce computational cost and memory usage.

## 6 EXPERIMENTS

### 6.1 PRETRAINING

To investigate the performance of the representation based on weighted point sets, Weighted Point Set Embedding (WPSE), we conducted experiments in which we trained a text-image representation model. We utilized Conceptual Captions 3M (CC3M) (Sharma et al., 2018) and Conceptual Captions 12M (CC12M) (Changpinyo et al., 2021) as datasets for pretraining. As the base architecture of the image encoder, we adopted ViT-B/16 (Dosovitskiy et al., 2021). Following SLIP (Mu et al., 2022), we used the smallest text Transformer model from CLIP. We modified the image encoder

Table 1: Zero-shot classification performance. We report the mean per-class accuracy (%) on Caltech-101, Aircraft, Flowers, and Pets. On other datasets, we report the top-1 accuracy (%).

| | Model | Average | ImageNet | CIFAR-10 | CIFAR-100 | STL-10 | Food-101 | Caltech-101 | Cars | Aircraft | Flowers | EuroSAT | DTD | Pets | SUN397 |
|---|---|---|---|---|---|---|---|---|---|---|---|---|---|---|---|
| CC3M | CLIP | 25.03 | 19.94 | 59.25 | 22.48 | 75.24 | 13.05 | 47.20 | 1.11 | 1.38 | **13.11** | 10.40 | 13.56 | **14.62** | 34.06 |
| | WPSE Gaussian | 26.75 | 21.20 | 59.95 | 23.58 | 80.61 | **14.56** | **51.18** | **1.49** | 1.35 | 12.60 | 19.98 | 13.40 | 13.60 | **34.16** |
| | WPSE IMQ | **27.04** | **21.36** | **61.22** | **25.91** | **81.64** | 13.17 | 50.15 | 1.41 | **1.84** | 12.14 | **22.02** | **13.69** | 13.88 | 33.05 |
| CC12M | CLIP | 43.78 | 39.15 | 74.17 | 42.98 | 90.91 | 47.96 | 73.58 | 21.94 | 2.01 | 29.71 | 22.24 | **22.45** | 52.29 | **49.72** |
| | WPSE Gaussian | **46.12** | **39.95** | **81.33** | **49.49** | 91.25 | 50.63 | **74.66** | **24.14** | **2.54** | **30.11** | 23.28 | 21.17 | **61.41** | 49.57 |
| | WPSE IMQ | 45.71 | 39.26 | 80.31 | 47.53 | **91.83** | **51.82** | 73.54 | 21.92 | 1.62 | 29.53 | **28.36** | 21.62 | 57.31 | 49.54 |

Table 2: Linear classification performance. We report the mean per-class accuracy (%) on Caltech-101, Aircraft, Flowers, and Pets. On other datasets, we report the top-1 accuracy (%).

| | Model | Average | ImageNet | CIFAR-10 | CIFAR-100 | STL-10 | Food-101 | Caltech-101 | Cars | Aircraft | Flowers | EuroSAT | DTD | Pets | SUN397 |
|---|---|---|---|---|---|---|---|---|---|---|---|---|---|---|---|
| CC3M | CLIP | 67.00 | 51.42 | 85.51 | 64.87 | 91.71 | 61.71 | 79.24 | 27.27 | 31.81 | 86.67 | 93.82 | 63.19 | 66.48 | 67.35 |
| | WPSE Gaussian | **69.01** | 56.18 | 85.00 | **65.10** | **92.20** | 63.71 | 79.97 | **30.68** | **37.85** | **88.63** | **94.94** | **64.15** | **69.08** | **69.64** |
| | WPSE IMQ | 68.23 | **56.77** | **85.87** | 63.48 | 92.06 | **64.12** | **80.78** | 27.16 | 33.98 | 87.47 | 93.72 | 63.14 | 69.06 | 69.35 |
| | CLIP (bef) | 72.14 | 58.33 | 87.90 | 70.05 | 92.64 | 66.07 | 82.49 | 39.46 | 44.96 | 91.48 | **96.02** | 67.02 | 71.86 | 69.56 |
| | WPSE Gaussian (bef) | 73.77 | **61.19** | 87.94 | **70.36** | **92.70** | **69.37** | 84.03 | **44.50** | **47.93** | **92.10** | 95.86 | **67.71** | 74.10 | **71.21** |
| | WPSE IMQ (bef) | **73.81** | 61.02 | **88.44** | 70.10 | 92.68 | 68.84 | **84.39** | 43.90 | 47.66 | 91.98 | 95.92 | 67.61 | **75.94** | 71.10 |
| CC12M | CLIP | 77.89 | 65.15 | 91.06 | 71.75 | 95.34 | 77.47 | 87.24 | 64.53 | 41.93 | 92.50 | 94.32 | **72.98** | 81.71 | 76.55 |
| | WPSE Gaussian | **79.08** | **67.83** | **91.72** | **73.06** | 96.46 | **79.49** | **89.18** | 65.23 | **44.53** | 92.09 | 94.58 | 72.93 | **83.56** | **77.42** |
| | WPSE IMQ | 78.90 | 67.11 | 91.14 | 72.59 | **96.48** | 78.69 | 88.85 | **66.32** | 44.17 | **92.61** | **94.70** | 72.55 | 83.21 | 77.31 |
| | CLIP (bef) | 81.03 | 69.15 | 92.04 | 75.99 | 95.40 | 80.13 | 90.09 | 70.86 | 53.72 | 94.85 | **96.64** | 74.89 | 81.81 | 77.75 |
| | WPSE Gaussian (bef) | 82.52 | **70.94** | 93.00 | 77.16 | 96.53 | **81.76** | **91.65** | 74.28 | 55.61 | 95.22 | 96.30 | **76.22** | 85.51 | **78.62** |
| | WPSE IMQ (bef) | **82.71** | 70.90 | **93.04** | **77.27** | **96.60** | 81.58 | 91.06 | **75.53** | **58.10** | **95.60** | 96.28 | 75.43 | **85.65** | 78.19 |

and the text encoder to produce weighted point sets (as explained in Section 5.3). As a nonlinear kernel $\tilde{k}$, we used the Gaussian kernel and the IMQ kernel. We performed hyperparameter search over $\sigma$ (for the Gaussian kernel) and $c$ (for the IMQ kernel) in the range of $\{0.5, 0.75, 1.0\}$. We also ran a hyperparameter search on the coefficients $(\alpha_1, \alpha_2)$ for combination kernels. We searched $(\alpha_1, \alpha_2) = (0.667, 0.333), (0.6, 0.4), (0.5, 0.5), (0.4, 0.6), (0.333, 0.667)$. In the tables, we report the performance of the best model from the hyperparameter search. During the pretraining, we set the dimension $D$ of RFFs to 1024. For each batch, new $\omega_t$ and $\beta_t$ for RFFs were sampled during the pretraining. For comparison, we also trained typical CLIP models from scratch. For more training details, see Appendix A.2.

## 6.2 ZERO-SHOT TRANSFER

We evaluated the zero-shot classification performance on the following 13 benchmark datasets: ImageNet (Russakovsky et al., 2015), CIFAR-10 (Krizhevsky, 2009), CIFAR-100 (Krizhevsky, 2009), STL-10 (Coates et al., 2011), Food-101 (Bossard et al., 2014), Caltech-101 (Fei-Fei et al., 2006), Stanford Cars (Krause et al., 2013), FGVC Aircraft (Maji et al., 2013), Oxford Flowers (Nilsback & Zisserman, 2008), EuroSAT (Helber et al., 2019), Describable Textures Dataset (DTD) (Cimpoi et al., 2014), Oxford Pets (Parkhi et al., 2012), and SUN397 (Xiao et al., 2010). Following SLIP (Mu et al., 2022), we adopted prompt ensembling and utilized prompts provided by SLIP for each dataset. We set the dimension $D$ of RFFs to 512. $\omega_t$ and $\beta_t$ for RFFs were fixed before the evaluation. To investigate the effect of the randomness of RFFs, we performed five evaluations for the models that use RFFs. Table 1 lists the zero-shot classification results, where the results of models using RFFs have been averaged. Additionally, Table 5 in the Appendix shows the standard deviation. As these findings show, the proposed method outperformed CLIP on average. In addition, the randomness of RFFs did not have a significant impact on the overall performance.

## 6.3 LINEAR CLASSIFICATION

We also performed the linear classification evaluation where we trained linear classifiers on the embedding vectors obtained by frozen pretrained image encoders. We used the same 13 benchmarks as the zero-shot classification. To extract embeddings for training linear classifiers, we used two different settings. In the first setting, we used the embeddings that were used for computation of the similarity in the symmetric InfoNCE. We set $D$ for RFFs to 512. $\omega_t$ and $\beta_t$ were fixed before the evaluation. Based on the robustness of the RFFs shown in Table 5, we did not evaluate multiple settings of $\omega_t$ and $\beta_t$. In the second setting, following common practice (Chen et al., 2021), we used the intermediate latent vectors just before the last projection layer of the image encoder. We denote this setting as "(bef)" in tables. For our WPSE models, weighted sum of the latent vectors with weights in the output weighted point set was used, and no RFF was used.

We basically followed the evaluation procedure in Fürst et al. (2022). We used a logistic regression classifier with an L-BFGS optimizer (Liu & Nocedal, 1989) and the maximum number of iteration of 1000. We utilized the implementation from cuML (Raschka et al., 2020). For hyperparameter tuning of the L2 regularization cost, we followed the protocol of CLIP (Radford et al., 2021). We ran hyperparameter sweeps over $C \in [10^{-6}, 10^6]$ with a parametric binary search on a validation split of each dataset. For datasets that do not provide an official validation split, we randomly split the training dataset into training and validation splits. After the hyperparameter was determined, we trained a classifier on the combination of training and validation splits and report its performance on the test split. Table 2 lists the linear classification results. Overall, our proposed method outperformed CLIP on average.

## 6.4 ABLATION STUDY

To investigate the effectiveness of our similarity, we trained two variant models that output weighted point sets on CC3M. One model outputs weighted point sets but the all weights are positive: $w_i \geq 0$. We used a function $100\mathrm{Sigmoid}(\cdot/100)$ as the last activation for weights in encoders. We denote this model as WPSE with positive weights. The other model also outputs weighted point sets but the similarity of weighted point sets are calculated only

Table 3: Ablation study. Models are trained on CC3M. Except for WPSE Linear, IMQ kernel was used.

| Model | Zero-shot | Linear |
|---|---|---|
| WPSE | 27.04 | 68.23 |
| WPSE with positive weights | 4.22 | – |
| WPSE Linear | 27.25 | 67.40 |

with linear kernel, i.e., the coefficients $(\alpha_1, \alpha_2)$ are set to $(1, 0)$. We denote this model as WPSE Linear. We also trained the model with the coefficients $(\alpha_1, \alpha_2) = (0, 1)$, specifically only with the nonlinear kernel. However, the training of this model failed due to a NaN loss error. Table 3 shows the average performance of zero-shot classification and linear classification on the 13 benchmark datasets. For WPSE with positive weights, we used the same parameters for the combination kernel. This indicates that negative weights are crucial for a good performance. (We did not perform linear classifications for WPSE with positive weights.) In comparison to WPSE Linear, it indicates the superiority of the use of non-linear kernel in the linear classification tasks. We also show the result of ablation study using CC12M in Appendix A.4.

## 7 CONCLUSION

We proposed a multimodal representation learning with weighted point sets. In our method, each input is transformed by an encoder into a weighted point set representation. The similarity between two weighted point sets is calculated with a kernel function that defines the similarity of two points. We also showed the theoretical benefits of using our representation and similarity. We highlighted that the optimal similarity of the symmetric InfoNCE is represented by the pointwise mutual information and showed that we can construct a linear classifier close to the optimal classifier of downstream tasks that is possibly nonlinear when the optimal similarity is obtained. In addition, we clarified the effect on the performance of downstream tasks caused by the deviation of the obtained similarity from the pointwise mutual information, and explained that the deviation of the similarity can be suppressed when using the proposed similarity based on weighted point sets. Experiments on text-image datasets demonstrated the superior performance of the proposed method compared to baselines.

ETHICS STATEMENT

In conducting this research on representation learning models, we are committed to upholding ethical standards. Our work aims to contribute to machine learning research society by theoretical analysis of representation learning and enhancing the capability of representations. However, we recognize potential concerns of representation learning models, such as biases in training datasets, license issues of scraped datasets and harmful applications. We acknowledge that representation learning models can have significant impacts on society. Therefore, we commit ourselves to ensuring that our research activity positively contributes to society while avoiding harm.

REPRODUCIBILITY STATEMENT

Detailed descriptions of our setup of the algorithm and experiments can be found in Section 5.3, 6, and Appendix A. Moreover, we release our code at `https://github.com/sony/wpse` to ensure reproducibility.

ACKNOWLEDGEMENTS

Computational resource of AI Bridging Cloud Infrastructure (ABCI) provided by National Institute of Advanced Industrial Science and Technology (AIST) was used. TS was partially supported by JSPS KAKENHI (20H00576) and JST CREST (JPMJCR2015). We would like to extend our thanks to Wei-Hsiang Liao and Bac Nguyen from Sony AI for the valuable feedback.

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

---

**Algorithm 1** Symmetric InfoNCE loss with the similarity of weighted point sets

---

**Require:** an image encoder $f_{\mathcal{X}}$, a text encoder $f_{\mathcal{Y}}$, a batch of $B$ paired images and texts $\{(x_b, y_b)\}_{b=1}^{B}$, the distribution $p_\omega$ associated with the shift-invariant kernel $\tilde{k}$, coefficients $\alpha_1$ and $\alpha_2$, and a temperature $\tau$.

1: $\left\{ \left( w_{b1}^{(\mathcal{X})}, v_{b1}^{(\mathcal{X})} \right), \ldots, \left( w_{bM^{(\mathcal{X})}}^{(\mathcal{X})}, v_{bM^{(\mathcal{X})}}^{(\mathcal{X})} \right) \right\} \leftarrow f_{\mathcal{X}}(x_b)$ for each $b \in [B]$

2: $\left\{ \left( w_{b1}^{(\mathcal{Y})}, v_{b1}^{(\mathcal{Y})} \right), \ldots, \left( w_{bM^{(\mathcal{Y})}}^{(\mathcal{Y})}, v_{bM^{(\mathcal{Y})}}^{(\mathcal{Y})} \right) \right\} \leftarrow f_{\mathcal{Y}}(y_b)$ for each $b \in [B]$

3: $\overline{v}_b^{(\mathcal{X})} \leftarrow \sum_{i=1}^{M^{(\mathcal{X})}} w_{bi}^{(\mathcal{X})} v_{bi}^{(\mathcal{X})}$

4: $\overline{v}_b^{(\mathcal{Y})} \leftarrow \sum_{j=1}^{M^{(\mathcal{Y})}} w_{bj}^{(\mathcal{Y})} v_{bj}^{(\mathcal{Y})}$

5: Draw $D$ i.i.d. samples $\omega_1, \ldots, \omega_D$ from $p_\omega$.

6: Draw $D$ i.i.d. samples $\beta_1, \ldots, \beta_D$ from $\mathrm{Unif}[0, 2\pi)$.

7: $\overline{z}_b^{(\mathcal{X})} \leftarrow \sum_{i=1}^{M^{(\mathcal{X})}} w_{bi}^{(\mathcal{X})} z \left( v_{bi}^{(\mathcal{X})}; \{\omega_t\}_{t=1}^{D}, \{\beta_t\}_{t=1}^{D} \right)$ for each $b \in [B]$

8: $\overline{z}_b^{(\mathcal{Y})} \leftarrow \sum_{j=1}^{M^{(\mathcal{Y})}} w_{bj}^{(\mathcal{Y})} z \left( v_{bj}^{(\mathcal{Y})}; \{\omega_t\}_{t=1}^{D}, \{\beta_t\}_{t=1}^{D} \right)$ for each $b \in [B]$

9: $S_{bb'} \leftarrow \tau^{-1} \left( \alpha_1 \overline{v}_b^{(\mathcal{X})\top} \overline{v}_{b'}^{(\mathcal{Y})} + \alpha_2 \overline{z}_b^{(\mathcal{X})\top} \overline{z}_{b'}^{(\mathcal{Y})} \right)$ for each $b, b' \in [B]$

10: Compute the symmetric InfoNCE loss from the similarity matrix $\{S_{bb'}\}_{bb'}$.

---

# A  ADDITIONAL DETAILS OF IMPLEMENTATION AND EXPERIMENTS

## A.1  IMPLEMENTATION

**Random Fourier feature (RFF)**   Random Fourier feature (Rahimi & Recht, 2007) is a technique for reducing computational complexity of kernel methods. For a shift-invariant kernel $k(u, v) = k(u - v)$ on $\mathbb{R}^d$ such that $k(0) = 1$, there exists a probability distribution, $p_\omega$, of a random variable, $\omega \in \mathbb{R}^d$ that satisfies:

$$k(u - v) = \mathop{\mathbb{E}}_{\omega, \beta} \left[ 2 \cos(\omega^\top u + \beta) \cos(\omega^\top v + \beta) \right],$$

where $\beta \in \mathbb{R}$ is sampled from a uniform distribution, $\mathrm{Unif}[0, 2\pi]$, over $[0, 2\pi]$. $p_\omega$ is given by the Fourier transform of $k(u - v)$. Based on this fact, we can construct an unbiased estimator of $k(u, v)$ as follows. First, $\omega_t \in \mathbb{R}^d$ and $\beta_t \in \mathbb{R}$ ($t = 1, \cdots, D$) are independently sampled from the distributions $p_\omega$ and $\mathrm{Unif}[0, 2\pi]$, respectively. Then, a vector $z(v) \in \mathbb{R}^D$ is constructed from $v \in \mathbb{R}^d$, $\{\omega_t\}_{t=1}^{D}$, and $\{\beta_t\}_{t=1}^{D}$ as

$$z \left( v; \{\omega_t\}_{t=1}^{D}, \{\beta_t\}_{t=1}^{D} \right) = \sqrt{\frac{2}{D}} \left[ \cos\left(\omega_1^\top v + \beta_1\right), \ldots, \cos\left(\omega_D^\top v + \beta_D\right) \right]^\top. \tag{9}$$

Similarly, $z(u)$ is constructed from $u$ with the same $\{\omega_t\}_{t=1}^{D}$ and $\{\beta_t\}_{t=1}^{D}$. Last, an estimator of $k(u, v)$ is obtained by taking the inner product of the vectors: $\mathbb{E}\left[z(u)^\top z(v)\right] = k(u, v)$. For the specific form of $p_\omega$ for Gaussian kernel and IMQ kernel, and further details, see Appendix C in Li et al. (2021). Algorithm 1 shows a pseudocode for computing our proposed similarity for symmetric InfoNCE.

**Model architecture**   In addition to the modifications in Section 5.3, we modify Transformer encoders as follows. To stabilize training, we add an activation function, $100 \tanh(\cdot / 100)$, after the projection layer for weights for restricting the range of weights. In preliminary experiments, we found that model parameters diverged during pretraining without it. Following CLIP, we apply L2-nomalization to points $v_i$ ($i \in [M]$) in weighted point sets, and use an inverse temperature parameter $\tau^{-1}$ to scale the similarity of weighted point sets (Algorithm 1). In typical CLIP implementations, $\tau^{-1}$ is calculated by an exponential activation as $\tau^{-1} = \exp(\theta)$ with a learnable parameter $\theta$ and clipping to a certain range, such as $[1, 100]$. However, in preliminary experiments, we found that $\tau^{-1}$ increased rapidly to the maximum value in the beginning of pretraining when we use the exponential activation and the weighted point set similarity, and that it harmed model performance. Therefore, we remove the exponential activation and use $\tau^{-1} = \theta$ to scale the proposed similarity. The range for clipping is set to $[1, 100]$.

## A.2 PRETRAINING

The text Transformer model we used is a 12-layer 512-wide transformer with eight attention heads. We utilized a byte pair encoding (BPE) tokenizer with a vocabulary size of 49K and a maximum context length of 77. Based on the Transformer architectures, we set $M^{(\mathcal{X})}$, $M^{(\mathcal{Y})}$, and $d$ of the weighted point sets to 197, 77, and 512, respectively. As a data augmentation, images were randomly resized and cropped with a scaling factor between 0.5 and 1.0 and bicubic interpolation. Models were trained for 50 epochs on CC3M and for 35 epochs on CC12M. We set the batch size to 2048. We used the AdamW optimizer with a beta2 of 0.98 and cosine scheduling with a linear warmup in pretraining. We set the initial learning rate to 0.0005 and used weight decay of 0.5. We used the built-in automatic mixed precision library in PyTorch (Paszke et al., 2019).

## A.3 CLASSIFICATION EVALUATIONS

Table 4: 13 datasets used for classification evaluations.

| Dataset | Classes | Train | Val | Test |
|---|---|---|---|---|
| ImageNet (Russakovsky et al., 2015) | 1000 | 1153051 | 128116 | 50000 |
| CIFAR-10 (Krizhevsky, 2009) | 10 | 45000 | 5000 | 10000 |
| CIFAR-100 (Krizhevsky, 2009) | 100 | 45000 | 5000 | 10000 |
| STL-10 (Coates et al., 2011) | 10 | 4500 | 500 | 8000 |
| Food-101 (Bossard et al., 2014) | 101 | 68175 | 7575 | 25250 |
| Caltech-101 (Fei-Fei et al., 2006) | 102 | 2754 | 306 | 6085 |
| Stanford Cars (Krause et al., 2013) | 196 | 7330 | 814 | 8041 |
| FGVC Aircraft (Maji et al., 2013) | 100 | 3334 | 3333 | 3333 |
| Oxford Flowers (Nilsback & Zisserman, 2008) | 102 | 1020 | 1020 | 6149 |
| EuroSAT (Helber et al., 2019) | 10 | 9000 | 1000 | 5000 |
| DTD (Cimpoi et al., 2014) | 47 | 1880 | 1880 | 1880 |
| Oxford Pets (Parkhi et al., 2012) | 37 | 3312 | 368 | 3669 |
| SUN397 (Xiao et al., 2010) | 397 | 76129 | 10867 | 21758 |

The properties of the datasets we used in the classification tasks are listed in Table 4. In Table 5, we show the results of the same zero-shot classification as presented in Section 6.2 but with the standard deviation included.

Table 5: Zero-shot classification performance.

| | CC3M | | CC12M | |
|---|---|---|---|---|
| Dataset | WPSE Gaussian | WPSE IMQ | WPSE Gaussian | WPSE IMQ |
| ImageNet | $21.20 \pm 0.05$ | $21.36 \pm 0.04$ | $39.95 \pm 0.06$ | $39.26 \pm 0.06$ |
| CIFAR-10 | $59.95 \pm 0.20$ | $61.22 \pm 0.51$ | $81.33 \pm 0.33$ | $80.31 \pm 0.24$ |
| CIFAR-100 | $23.58 \pm 0.13$ | $25.91 \pm 0.19$ | $49.49 \pm 0.16$ | $47.53 \pm 0.05$ |
| STL-10 | $80.61 \pm 0.37$ | $81.64 \pm 0.25$ | $91.25 \pm 0.09$ | $91.83 \pm 0.13$ |
| Food-101 | $14.56 \pm 0.08$ | $13.17 \pm 0.05$ | $50.63 \pm 0.08$ | $51.82 \pm 0.20$ |
| Caltech-101 | $51.18 \pm 0.12$ | $50.15 \pm 0.10$ | $74.66 \pm 0.20$ | $73.54 \pm 0.26$ |
| Cars | $1.49 \pm 0.02$ | $1.41 \pm 0.08$ | $24.14 \pm 0.16$ | $21.92 \pm 0.13$ |
| Aircraft | $1.35 \pm 0.12$ | $1.84 \pm 0.13$ | $2.54 \pm 0.09$ | $1.62 \pm 0.15$ |
| Flowers | $12.60 \pm 0.10$ | $12.14 \pm 0.15$ | $30.11 \pm 0.25$ | $29.53 \pm 0.26$ |
| EuroSAT | $19.98 \pm 0.18$ | $22.02 \pm 0.92$ | $23.28 \pm 0.36$ | $28.36 \pm 0.38$ |
| DTD | $13.40 \pm 0.24$ | $13.69 \pm 0.13$ | $21.17 \pm 0.23$ | $21.62 \pm 0.28$ |
| Pets | $13.60 \pm 0.18$ | $13.88 \pm 0.16$ | $61.41 \pm 0.15$ | $57.31 \pm 1.61$ |
| SUN397 | $34.16 \pm 0.11$ | $33.05 \pm 0.15$ | $49.57 \pm 0.11$ | $49.54 \pm 0.29$ |

## A.4 ABLATION STUDY ON CC12M

In this section, we present the result of ablation study using CC12M. We trained two variant models that output weighted point sets. One model is a WPSE Linear, which we described in Section 6.4, with the coefficients $(\alpha_1, \alpha_2) = (1, 0)$ and the similarity of weighted point sets are calculated only with linear kernel. The other model has the coefficients $(\alpha_1, \alpha_2) = (0, 1)$ and calculates the similarity of weighted point sets using only a nonlinear kernel. We denote this

Table 6: Ablation study. Models are trained on CC12M. Except for WPSE Linear, Gaussian kernel was used.

| Model | Zero-shot | Linear |
|---|---|---|
| WPSE | 46.12 | 79.08 |
| WPSE Nonlinear | 44.05 | 70.99 |
| WPSE Linear | 45.87 | 78.61 |

model as WPSE Nonlinear. Additionally, we also trained a WPSE with postive weights with the last sigmoid activation in the same manner as described inSection 6.4. However, the training of this model failed due to a NaN loss. Table 6 shows the average performance of zero-shot classification and linear classification on the 13 benchmark dataset. This indicates that the combination of the linear kernel and a nonlinear kernel is beneficial for the performance.

# B    PROOFS OF STATEMENTS IN SECTION 4

## B.1    PROOF OF THEOREM 4.2

*Proof.* From the definition of $\bar{h}^g(x)$, the $c$-th entry of $\bar{h}^{g^*}(x)$ is calculated as follows:

$$
\begin{aligned}
\bar{h}^{g^*}(x)_c &= \left( \mathop{\mathbb{E}}_{p(y|\mathcal{Y}_c)} \left[ \frac{1}{\tau^*} f_{\mathcal{Y}}^*(y) \right] \right)^\top f_{\mathcal{X}}^*(x) + \ln P(\mathcal{Y}_c) \\
&= \mathop{\mathbb{E}}_{p(y|\mathcal{Y}_c)} \left[ \frac{1}{\tau^*} f_{\mathcal{Y}}^*(y)^\top f_{\mathcal{X}}^*(x) \right] + \ln P(\mathcal{Y}_c) \\
&= \mathop{\mathbb{E}}_{p(y|\mathcal{Y}_c)} \left[ g^*(x, y) \right] + \ln P(\mathcal{Y}_c) \\
&= \mathop{\mathbb{E}}_{p(y|\mathcal{Y}_c)} \left[ \ln \frac{p(x, y)}{p(x)p(y)} \right] + \ln P(\mathcal{Y}_c) + \Gamma.
\end{aligned}
$$

Since adding a constant to all entries of $h(x)$ doesn't change the supervised loss $\mathcal{L}_{\mathrm{sup}}(h)$, we consider $\Gamma = 0$ for the sake of simplicity. The $c$-th entry of $\bar{h}^{g^*}(x)$ is further rearranged as follows:

$$
\begin{aligned}
\bar{h}^{g^*}(x)_c &= \mathop{\mathbb{E}}_{p(y|\mathcal{Y}_c)} \left[ \ln \frac{p(x, y)}{p(x)p(y)} \right] + \ln P(\mathcal{Y}_c) \\
&= \mathop{\mathbb{E}}_{p(y|\mathcal{Y}_c)} \left[ \ln \frac{p(x, y)p(x)P(\mathcal{Y}_c)}{p(x)p(y)p(x, \mathcal{Y}_c)} + \ln \frac{p(x, \mathcal{Y}_c)}{p(x)P(\mathcal{Y}_c)} \right] + \ln P(\mathcal{Y}_c) \\
&= \mathop{\mathbb{E}}_{p(y|\mathcal{Y}_c)} \left[ \ln \frac{p(x, y)/p(x, \mathcal{Y}_c)}{p(y)/P(\mathcal{Y}_c)} \right] + \ln \frac{p(x, \mathcal{Y}_c)}{p(x)} \\
&= \mathop{\mathbb{E}}_{p(y|\mathcal{Y}_c)} \left[ \ln \frac{p(y|x, \mathcal{Y}_c)}{p(y|\mathcal{Y}_c)} \right] + \ln P(\mathcal{Y}_c|x) \\
&= \ln P(\mathcal{Y}_c|x) - D_{\mathrm{KL}} \left( p_Y(Y|\mathcal{Y}_c) \parallel p_Y(Y|x, \mathcal{Y}_c) \right).
\end{aligned}
$$

Therefore, we have

$$
\begin{aligned}
& \mathcal{L}_{\mathrm{sup}}(\bar{h}^{g*}) - \mathcal{L}_{\mathrm{sup}}(h^*) \\
& = \underset{p(x,c)}{\mathbb{E}} \left[ \ln P(c|x) - \bar{h}^{g*}(x)_c + \ln\left( \sum_i \exp \bar{h}^{g*}(x)_i \right) \right] \\
& = \underset{p(x,c)}{\mathbb{E}} \left[ \ln P(c|x) - \ln P(\mathcal{Y}_c|x) + D_{\mathrm{KL}}\left( p_Y(Y|\mathcal{Y}_c) \,\|\, p_Y(Y|x,\mathcal{Y}_c) \right) \right. \\
& \qquad\qquad \left. + \ln\left( \sum_i P(\mathcal{Y}_i|x) \cdot \exp(-D_{\mathrm{KL}}\left( p_Y(Y|\mathcal{Y}_i) \,\|\, p_Y(Y|x,\mathcal{Y}_i) \right)) \right) \right] \\
& \leq \underset{p(x,c)}{\mathbb{E}} \left[ \ln P(c|x) - \ln P(\mathcal{Y}_c|x) + D_{\mathrm{KL}}\left( p_Y(Y|\mathcal{Y}_c) \,\|\, p_Y(Y|x,\mathcal{Y}_c) \right) + \ln\left( \sum_i P(\mathcal{Y}_i|x) \right) \right] \\
& = \underset{p(x,c)}{\mathbb{E}} \left[ \ln p(c|x) - \ln p(\mathcal{Y}_c|x) + D_{\mathrm{KL}}\left( p_Y(Y|\mathcal{Y}_c) \,\|\, p_Y(Y|x,\mathcal{Y}_c) \right) + \ln P(\tilde{\mathcal{Y}}|x) \right] \\
& = \underset{p(x,c)}{\mathbb{E}} \left[ \ln \frac{P(c|x)}{P(\mathcal{Y}_c|x)/P(\tilde{\mathcal{Y}}|x)} + D_{\mathrm{KL}}\left( p_Y(Y|\mathcal{Y}_c) \,\|\, p_Y(Y|x,\mathcal{Y}_c) \right) \right] \\
& = \underset{p(x)}{\mathbb{E}} \left[ D_{\mathrm{KL}}\left( P_C(C|x) \,\Big\|\, P_C(C \mid x; (\mathcal{Y}_i)_{i\in[K]}) \right) \right] + \underset{p(x,c)}{\mathbb{E}} \left[ D_{\mathrm{KL}}\left( p_Y(Y|\mathcal{Y}_c) \,\|\, p_Y(Y|x,\mathcal{Y}_c) \right) \right].
\end{aligned}
$$

Here, the inequality holds by the monotonicity of $\ln(\cdot)$, the non-negativity of $P(\mathcal{Y}_i|x)$, and the non-negativity of KL divergence. $\qquad\square$

## B.2 Proof of Lemma 4.3

*Proof.* For every $i \in [K]$, it holds that

$$
\begin{aligned}
\left| \bar{h}^g(x)_i - \bar{h}^{g^*}(x)_i \right| &= \left| \underset{p(y|\mathcal{Y}_i)}{\mathbb{E}} [g(x,y) - g^*(x,y)] \right| \\
&\leq \left| \underset{p(y|\mathcal{Y}_i)}{\mathbb{E}} [\Delta] \right| \\
&= \Delta.
\end{aligned}
$$

Let $\varsigma_c(z)$ denote the logarithm of the $c$-th entry of the softmax function, i.e., $\varsigma_c(z) := \ln \frac{e^{z_c}}{\sum_{i=1}^K e^{z_i}}$.

$$
\begin{aligned}
\left| \mathcal{L}_{\mathrm{sup}}(\bar{h}^g) - \mathcal{L}_{\mathrm{sup}}(\bar{h}^{g^*}) \right| &= \left| \underset{p(x,c)}{\mathbb{E}} \left[ -\ln \frac{\exp \bar{h}^g(x)_c}{\sum_{i=1}^K \exp \bar{h}^g(x)_i} + \ln \frac{\exp \bar{h}^{g^*}(x)_c}{\sum_{i=1}^K \exp \bar{h}^{g^*}(x)_i} \right] \right| \\
&\leq \underset{p(x,c)}{\mathbb{E}} \left[ \left| -\varsigma_c\left( \bar{h}^g(x) \right) + \varsigma_c\left( \bar{h}^{g^*}(x) \right) \right| \right] \qquad (10)
\end{aligned}
$$

$\varsigma_c(z)$ is a differentiable function with respect to $z$, and the partial derivative is given as follows:

$$
\begin{aligned}
\frac{\partial \varsigma_c}{\partial z_c} &= 1 - \frac{e^{z_c}}{\sum_{i=1}^K e^{z_i}}, \\
\frac{\partial \varsigma_c}{\partial z_j} &= \frac{-e^{z_j}}{\sum_{i=1}^K e^{z_i}} \quad \text{for } j \neq c.
\end{aligned}
$$

By the mean value theorem, there exists $\xi$ on the line segment between $\bar{h}^g(x)$ and $\bar{h}^{g^*}(x)$ such that

$$
-\varsigma_c\left( \bar{h}^g(x) \right) + \varsigma_c\left( \bar{h}^{g^*}(x) \right) = \nabla \varsigma_c(\xi)^\top \left( -\bar{h}^g(x) + \bar{h}^{g^*}(x) \right).
$$

Therefore, we have

$$
\begin{aligned}
\mathbb{E}_{p(x,c)}\left[\left|-\varsigma_c\left(\bar{h}^g(x)\right) + \varsigma_c\left(\bar{h}^{g^*}(x)\right)\right|\right] &= \mathbb{E}_{p(x,c)}\left[\left|\nabla\varsigma_c(\xi)^\top\left(-\bar{h}^g(x) + \bar{h}^{g^*}(x)\right)\right|\right] \\
&\leq \mathbb{E}_{p(x,c)}\left[\left(\sum_{i=1}^K\left|\frac{\partial\varsigma_c}{\partial z_i}(\xi)\right|\right)\left\|\bar{h}^g(x) - \bar{h}^{g^*}(x)\right\|_\infty\right] \\
&\leq \mathbb{E}_{p(x,c)}\left[2\Delta\right] \\
&= 2\Delta.
\end{aligned}
\tag{11}
$$

Here, the first inequality holds by Hölder's inequality. At the second inequality, we use

$$
\sum_{i=1}^K\left|\frac{\partial\varsigma_c}{\partial z_i}(\xi)\right| = 1 - \frac{e^{\xi_c}}{\sum_{i=1}^K e^{\xi_i}} + \frac{\sum_{i\neq c}e^{\xi_i}}{\sum_{i=1}^K e^{\xi_i}} \leq 2.
$$

Combining Eq. 10, 11 finishes the proof. $\qquad\square$

## C PROOFS OF STATEMENTS IN SECTION 5

### C.1 LIMITATION OF THE BILINEAR SIMILARITY

**Proposition C.1.** *Let $A, B \in \mathbb{R}^{d\times M}$, and $c \in \mathbb{R}$. Let $J \in \mathbb{R}^{M\times M}$ denote the matrix in which all entries are 1. Then, we have $\mathrm{rank}(A^\top B - cJ) \leq d + 1$.*

*Proof.* We define $\tilde{A}, \tilde{B} \in \mathbb{R}^{(d+1)\times M}$ as follows:

$$
\tilde{A} = \left[\begin{array}{c} A \\ \hline -1 \quad \cdots \quad -1 \end{array}\right], \quad \tilde{B} = \left[\begin{array}{c} B \\ \hline c \quad \cdots \quad c \end{array}\right].
$$

Then, we have $\tilde{A}^\top\tilde{B} = A^\top B - cJ$. Since $\mathrm{rank}\,\tilde{A} \leq d + 1$ and $\mathrm{rank}\,\tilde{B} \leq d + 1$, the statement holds. $\qquad\square$

### C.2 REPRESENTATIONAL CAPABILITY OF THE SIMILARITY BETWEEN WEIGHTED POINT SETS

We denote (joint) probability density functions of random variables by using their corresponding letters. For example, we denote the joint probability density function of the random variables $\tilde{X}, \tilde{Y}$ and the probability density function of $\tilde{X}$ as $p_{\tilde{X},\tilde{Y}}$ and $p_{\tilde{X}}$, respectively.

We impose the following assumptions on the generation process of random variables $X \in \mathcal{X}$ and $Y \in \mathcal{Y}$.

**Assumption C.2** (Generation process)**.** There exist random variables $\tilde{X}, \tilde{Y} \in \mathbb{R}^d$, $Z^{(\mathcal{X})} \in \mathbb{R}^{d_\mathcal{X}}$ and $Z^{(\mathcal{Y})} \in \mathbb{R}^{d_\mathcal{Y}}$ that satisfy the following conditions.

(a) $(\tilde{X}, \tilde{Y})$, $Z^{(\mathcal{X})}$, and $Z^{(\mathcal{Y})}$ are mutually independent.

(b) There exist continuous bijective mappings $h_\mathcal{X}\colon \mathbb{R}^d \times \mathbb{R}^{d_\mathcal{X}} \to \mathcal{X}$ and $h_\mathcal{Y}\colon \mathbb{R}^d \times \mathbb{R}^{d_\mathcal{Y}} \to \mathcal{Y}$ such that $X = h_\mathcal{X}(\tilde{X}, Z^{(\mathcal{X})})$ and $Y = h_\mathcal{Y}(\tilde{Y}, Z^{(\mathcal{Y})})$.

(c) The support $\mathrm{supp}\,p_{\tilde{X},\tilde{Y}} \subseteq \mathbb{R}^d \times \mathbb{R}^d$ of the distribution $p_{\tilde{X},\tilde{Y}}$ is compact.

(d) The pointwise mutual information $\mathrm{PMI}_{\tilde{X},\tilde{Y}}(\tilde{x},\tilde{y}) := \ln\frac{p_{\tilde{X},\tilde{Y}}(\tilde{x},\tilde{y})}{p_{\tilde{X}}(\tilde{x})p_{\tilde{Y}}(\tilde{y})}$ of $\tilde{X}$ and $\tilde{Y}$ is an $L$-Lipschitz function on $\mathrm{supp}\,p_{\tilde{X}} \times \mathrm{supp}\,p_{\tilde{Y}}$.

The second assumption means that data samples, $X$ and $Y$, are generated from low-dimensional latent variables, $(\tilde{X}, Z^{(\mathcal{X})})$ and $(\tilde{Y}, Z^{(\mathcal{Y})})$, respectively. The first assumption means that dependency between $X$ and $Y$ stems only from $\tilde{X}$ and $\tilde{Y}$, and that $Z^{(\mathcal{X})}$ and $Z^{(\mathcal{Y})}$ are latent variables specific to the domain $\mathcal{X}$ and $\mathcal{Y}$, respectively. From the first and second assumptions, it follows that there exists

a 1-to-1 correspondence between $(x, y) \in \mathcal{X} \times \mathcal{Y}$ and $(\tilde{x}, \tilde{y}, z^{(\mathcal{X})}, z^{(\mathcal{Y})}) \in \mathbb{R}^d \times \mathbb{R}^d \times \mathbb{R}^{d_{\mathcal{X}}} \times \mathbb{R}^{d_{\mathcal{Y}}}$, and that $\frac{p_{X,Y}(x,y)}{p_X(x)p_Y(y)} = \frac{p_{\tilde{X},\tilde{Y}}(\tilde{x},\tilde{y})p_{Z(\mathcal{X})}(z^{(\mathcal{X})})p_{Z(\mathcal{Y})}(z^{(\mathcal{Y})})}{p_{\tilde{X}}(\tilde{x})p_{Z(\mathcal{X})}(z^{(\mathcal{X})})p_{\tilde{Y}}(\tilde{y})p_{Z(\mathcal{Y})}(z^{(\mathcal{Y})})} = \frac{p_{\tilde{X},\tilde{Y}}(\tilde{x},\tilde{y})}{p_{\tilde{X}}(\tilde{x})p_{\tilde{Y}}(\tilde{y})}$.

To prove Theorem 5.1, we use the following statements.

**Proposition C.3** ((Aronszajn, 1950; Sriperumbudur et al., 2011)). *Let $X$ be a topological space and let $\mathcal{H}$ be a reproducing kernel Hilbert space of the functions on $X$ with $k \colon X \times X \to \mathbb{R}$ as its reproducing kernel. Then,*

$$\left\{ \sum_{j \in [n]} c_j k(\cdot, x_j) \;\middle|\; n \in \mathbb{N}, \{c_j : j \in [n]\} \subset \mathbb{R}, \{x_j : j \in [n]\} \subset X \right\}$$

*is dense in $\mathcal{H}$.*

**Lemma C.4.** *Let $X$ be a topological space and let $\mathcal{H}$ be a reproducing kernel Hilbert space of the functions on $X$ with a bounded kernel $k \colon X \times X \to \mathbb{R}$. Let $\sup_{x \in X} k(x, x) \leq \kappa$. For any $f, g \in \mathcal{H}$, if $\|f - g\|_{\mathcal{H}} < \varepsilon$, then $\|f - g\|_{\infty} < \sqrt{\kappa}\varepsilon$.*

*Proof.* For any $x \in X$,
$$|f(x) - g(x)| = \langle k(x, \cdot), f - g \rangle_{\mathcal{H}} \leq \|k(x, \cdot)\|_{\mathcal{H}} \|f - g\|_{\mathcal{H}} < \sqrt{\kappa}\varepsilon.$$
$\square$

**Definition C.5** ($c_0$-universal, (Sriperumbudur et al., 2011)). A bounded kernel, $k$ with $k(\cdot, x) \in C_0(X), \forall x \in X$ on a locally compact Hausdorff space $X$, is said to be $c_0$-universal if the RKHS, $\mathcal{H}$ induced by $k$ is dense in $C_0(X)$ w.r.t. the uniform norm. I.e., for every function $g \in C_0(X)$ and all $\varepsilon > 0$, there exists an $f \in \mathcal{H}$ such that $\|f - g\|_{\infty} \leq \varepsilon$.

We now present the proof of Theorem 5.1.

*Proof of Theorem 5.1.* First, we fix $\varepsilon > 0$. We prove the statement by explicitly constructing $M^{(\mathcal{X})}, M^{(\mathcal{Y})}, f_{\mathcal{X}}$, and $f_{\mathcal{Y}}$ that satisfy Eq.7.

From (b) of Assumption C.2, there exist continuous inverse functions of $h_{\mathcal{X}}$ and $h_{\mathcal{Y}}$. Consider the following restrictions of the functions $h_{\mathcal{X}}^{-1}$ and $h_{\mathcal{Y}}^{-1}$: for $x = h_{\mathcal{X}}(\tilde{x}, z^{(\mathcal{X})})$ and $y = h_{\mathcal{Y}}(\tilde{y}, z^{(\mathcal{Y})})$, it holds that

$$\tilde{x} = h_{\mathcal{X}}^{-1}|_{\tilde{X}}(x),$$
$$z^{(\mathcal{X})} = h_{\mathcal{X}}^{-1}|_{Z^{(\mathcal{X})}}(x),$$
$$\tilde{y} = h_{\mathcal{Y}}^{-1}|_{\tilde{Y}}(y),$$
$$z^{(\mathcal{Y})} = h_{\mathcal{Y}}^{-1}|_{Z^{(\mathcal{Y})}}(y).$$

Then, from (a) of Assumption C.2, it follows that

$$\begin{aligned}
\frac{p_{X,Y}(x,y)}{p_X(x)p_Y(y)} &= \frac{p_{\tilde{X},\tilde{Y}}(\tilde{x},\tilde{y})p_{Z^{(\mathcal{X})}}(z^{(\mathcal{X})})p_{Z^{(\mathcal{Y})}}(z^{(\mathcal{Y})})}{p_{\tilde{X}}(\tilde{x})p_{Z^{(\mathcal{X})}}(z^{(\mathcal{X})})p_{\tilde{Y}}(\tilde{y})p_{Z^{(\mathcal{Y})}}(z^{(\mathcal{Y})})} \\
&= \frac{p_{\tilde{X},\tilde{Y}}(\tilde{x},\tilde{y})}{p_{\tilde{X}}(\tilde{x})p_{\tilde{Y}}(\tilde{y})} \\
&= \frac{p_{\tilde{X},\tilde{Y}}(h_{\mathcal{X}}^{-1}|_{\tilde{X}}(x), h_{\mathcal{Y}}^{-1}|_{\tilde{Y}}(y))}{p_{\tilde{X}}(h_{\mathcal{X}}^{-1}|_{\tilde{X}}(x))p_{\tilde{Y}}(h_{\mathcal{Y}}^{-1}|_{\tilde{Y}}(y))}.
\end{aligned} \tag{12}$$

To avoid complicated notations, we simply denote $h_{\mathcal{X}}^{-1}|_{\tilde{X}}(x)$ as $\tilde{x}(x)$ and $h_{\mathcal{Y}}^{-1}|_{\tilde{Y}}(y)$ as $\tilde{y}(y)$ in the following.

From (c) of Assumption C.2, Proposition C.3, Lemma C.4, and the definition of the $c_0$-universal kernel, for any fixed $\tilde{y} \in \operatorname{supp} p_{\tilde{Y}}$, there exist $M \in \mathbb{N}$, $\{c_j \in \mathbb{R} \mid j \in [M]\}$, and $\{\tilde{\eta}_j \in \mathbb{R}^d \mid j \in [M]\}$ such that, for any $\tilde{x} \in \operatorname{supp} p_{\tilde{X}}$,

$$\left| \mathrm{PMI}_{\tilde{X},\tilde{Y}}(\tilde{x}, \tilde{y}) - \sum_{j \in [M]} c_j k(\tilde{x}, \tilde{\eta}_j) \right| < \frac{\varepsilon}{2}. \tag{13}$$

We denote such $M, c_j$, and $\tilde{\eta}_j$ as $M(\tilde{y}), c_j(\tilde{y})$ and $\tilde{\eta}_j(\tilde{y})$, respectively.

Meanwhile, we define $B_r(\tilde{y}) \subset \mathbb{R}^d$ as the open ball of radius $r$ and center $\tilde{y} \in \mathbb{R}^d$. From (c) of Assumption C.2, the support of $p_{\tilde{Y}}$ is compact. Thus, for any $\varepsilon > 0$, there exist $J \in \mathbb{N}$ and $J$ points $\tilde{y}_1, \tilde{y}_2, \cdots, \tilde{y}_J \in \mathbb{R}^d$ such that $\operatorname{supp} p_{\tilde{Y}} \subseteq \bigcup_{j=1}^{J} B_{\varepsilon/(2L)}(\tilde{y}_j)$. Given such $\tilde{y}_j$ $(j \in [J])$, we define $\chi(\tilde{y})$ for $\tilde{y} \in S$ as one of the points, $\tilde{y}_j$ $(j \in [J])$ that satisfies $\tilde{y} \in B_{\varepsilon/(2L)}(\tilde{y}_j)$. From (d) of Assumption C.2, it holds that, for any $(\tilde{x}, \tilde{y}) \in \operatorname{supp} p_{\tilde{X}, \tilde{Y}}$,

$$\left| \operatorname{PMI}_{\tilde{X}, \tilde{Y}}(\tilde{x}, \tilde{y}) - \operatorname{PMI}_{\tilde{X}, \tilde{Y}}(\tilde{x}, \chi(\tilde{y})) \right| < \frac{\varepsilon}{2}. \tag{14}$$

Now, we are ready to construct desirable $M^{(\mathcal{X})}, M^{(\mathcal{Y})}, f_{\mathcal{X}}$ and $f_{\mathcal{Y}}$. Let $M^{(\mathcal{X})} = 1$ and $M^{(\mathcal{Y})} = \max_{j \in [J]} M(\tilde{y}_j)$. We define $f_{\mathcal{Y}} : y \mapsto \left\{ \left( w_j^{(\mathcal{Y})}, v_j^{(\mathcal{Y})} \right) \right\}_{j \in [M^{(\mathcal{Y})}]}$ as

$$\begin{aligned} w_j^{(\mathcal{Y})} &= c_j(\chi(\tilde{y}(y))) && \text{for } 1 \le j \le M(\chi(\tilde{y}(y))), \\ w_j^{(\mathcal{Y})} &= 0 && \text{for } M(\chi(\tilde{y}(y))) < j \le M^{(\mathcal{Y})}, \\ v_j^{(\mathcal{Y})} &= \tilde{\eta}_j(\chi(\tilde{y}(y))) && \text{for } 1 \le j \le M(\chi(\tilde{y}(y))). \end{aligned}$$

For $v_j^{(\mathcal{Y})}$ with $j$ such that $M(\chi(\tilde{y}(y))) < j \le M^{(\mathcal{Y})}$, we can choose any point in $\mathbb{R}^d$. We define $f_{\mathcal{X}}$ as $f_{\mathcal{X}}(x) = \{(w_1, v_1)\} := \{(1, \tilde{x}(x))\}$. Then, for every $(x, y) \in \operatorname{supp} p_{X,Y} \subseteq \mathcal{X} \times \mathcal{Y}$,

$$\begin{aligned} &\left| \ln \frac{p_{X,Y}(x,y)}{p_X(x)p_Y(y)} - \tilde{g}(f_{\mathcal{X}}(x), f_{\mathcal{Y}}(y)) \right| \\ &= \left| \operatorname{PMI}_{\tilde{X}, \tilde{Y}}(\tilde{x}(x), \tilde{y}(y)) - \sum_{i=1}^{M^{(\mathcal{X})}} \sum_{j=1}^{M^{(\mathcal{Y})}} w_i^{(\mathcal{X})} w_j^{(\mathcal{Y})} k(v_i^{(\mathcal{X})}, v_j^{(\mathcal{Y})}) \right| \\ &\le \left| \operatorname{PMI}_{\tilde{X}, \tilde{Y}}(\tilde{x}(x), \tilde{y}(y)) - \operatorname{PMI}_{\tilde{X}, \tilde{Y}}(\tilde{x}(x), \chi(\tilde{y}(y))) \right| \\ &\quad + \left| \operatorname{PMI}_{\tilde{X}, \tilde{Y}}(\tilde{x}(x), \chi(\tilde{y}(y))) - \sum_{i=1}^{M^{(\mathcal{X})}} \sum_{j=1}^{M^{(\mathcal{Y})}} w_i^{(\mathcal{X})} w_j^{(\mathcal{Y})} k(v_i^{(\mathcal{X})}, v_j^{(\mathcal{Y})}) \right| \\ &\le \frac{\varepsilon}{2} + \left| \operatorname{PMI}_{\tilde{X}, \tilde{Y}}(\tilde{x}(x), \chi(\tilde{y}(y))) - \sum_{j=1}^{M(\chi(\tilde{y}(y)))} c_j\Big(\chi(\tilde{y}(y))\Big) k\Big(\tilde{x}(x), \tilde{\eta}_j(\chi(\tilde{y}(y)))\Big) \right| \\ &< \frac{\varepsilon}{2} + \frac{\varepsilon}{2} = \varepsilon. \end{aligned}$$

Here, the first inequality holds by the triangle inequality. The second inequality holds from Eq. 14 and the definitions of $f_{\mathcal{X}}$ and $f_{\mathcal{Y}}$. The third inequality holds from Eq. 13. $\qquad\square$

