# OpenReview forum: "Weighted Point Set Embedding for Multimodal Contrastive Learning Toward Optimal Similarity Metric"
_ICLR.cc/2025/Conference — ICLR 2025 Spotlight_

### Official Review · Reviewer_2vyS · 2024-11-02

**Soundness:** 3
**Presentation:** 3
**Contribution:** 3
**Rating:** 8
**Confidence:** 4

**Summary:**

The paper proposes to replace the point-wise contrastive loss of CLIP with a dense contrastive loss calculated by using the contrastive loss on top of the average pair-wise similarity between dense features (the features prior to the final pooling layer of encoders). The authors show that this leads to better results (using the same evaluation as CLIP). The method is rigorously motivated by analyzing excess risk relative to the optimal downstream classifier head.

**Strengths:**

(1) The introduction is especially well-written. I appreciated the motivation that many-to-many correspondences are not adequately represented by a single CLIP embedding.

(2) The theory is nice (but Eq. 3 has typos).

**Weaknesses:**

(1) It is unclear how this method differs from dense contrastive loses. (For example: https://openaccess.thecvf.com/content/CVPR2021/papers/Wang_Dense_Contrastive_Learning_for_Self-Supervised_Visual_Pre-Training_CVPR_2021_paper.pdf)

(2) Computational cost. The contrastive loss is already quite computationally expensive (square of mini-batch size if not using global hard negative mining). Now we multiply that cost by the square of number of feature vectors.

(3) Practically speaking (not considering the theory), there is not much innovation in this paper. Similarity metrics over pairs of sets of features are well-established. For example: ColBERT (https://arxiv.org/pdf/2004.12832). The extension from a pure NLP setting to image-text is straightforward.

**Questions:**

None.

---

> ### Author Response · Authors · 2024-11-24
> **Response to Reviewer 2vyS**
>
> Thank you very much for your thoughtful feedback and concerns. We appreciate your insights and would like to provide responses to your comments.
>
> > The theory is nice (but Eq. 3 has typos).
>
> We are grateful for your positive feedback and appreciate your pointing out the typo. We have revised our draft accordingly.
>
> > Computational cost. The contrastive loss is already quite computationally expensive (square of mini-batch size if not using global hard negative mining). Now we multiply that cost by the square of the number of feature vectors.
>
> Here, we would like to clarify that this computational cost of the product of the numbers of feature vectors, $M^{(\mathcal{X})} M^{(\mathcal{Y})}$, is avoided by utilizing the random Fourier feature (RFF) technique. Please refer to around Lines 377-402 for more details. With this RFF technique, the computational cost of obtaining a similarity matrix is reduced from $O(B^2M^{(\mathcal{X})}M^{(\mathcal{Y})}d)$ to $O(B^2D + BD(M^{(\mathcal{X})}+M^{(\mathcal{Y})}))$, where $B$ is the batch size, $d$ is the feature dimension, and $D$ is the RFF dimension. Moreover, these calculations mostly involve additions and multiplications of vectors, which can be performed quickly by GPUs.
>
> > It is unclear how this method differs from dense contrastive loss.
>
> > Practically speaking (not considering the theory), there is not much innovation in this paper. Similarity metrics over pairs of sets of features are well-established. For example: ColBERT (https://arxiv.org/pdf/2004.12832). The extension from a pure NLP setting to image-text is straightforward.
>
> * Ignoring minor differences, such as the methods for aggregating information from sets of (weighted) features and the use of maximum operations or non-linear kernels, we would like to highlight that our method offers a way to (approximately) aggregate information of a weighted point cloud into an embedding, as demonstrated around Eq. (8).
> Indeed, the dense contrastive loss aggregates the softmax cross-entropy objectives calculated from bags of embeddings, and ColBERT has explored similarity metrics that aggregate information of sets of features. However, to produce an aggregated embedding from a set of features that can be used for linear classification tasks, additional processes are required to transform the set of features. In contrast, our approach provides a method to aggregate information of a weighted point cloud into an embedding, which is theoretically grounded in kernel methods.
> On the other hand, we acknowledge that using a deep neural network that takes all features in the set as input may work and resolve this issue.
> * Setting aside theoretical aspects, there is room for investigation into more effective similarity metrics between pairs of sets of (weighted) features. As Reviewer Sno6 mentioned, considering concepts related to point clouds, such as neighborhoods and connectivity, would also be a promising and interesting direction.
> However, we would like to emphasize that our method has a theoretical guarantee in optimization of the symmetric InfoNCE and its capability of representing similarity matrices.

---

> > ### Comment · Reviewer_2vyS · 2024-11-25
> >
> > Thank you for the rebuttal. I read it and am confident with my initial score of accept.

---

> > > ### Author Response · Authors · 2024-11-26
> > >
> > > Thank you for your quick response. Your positive feedback has greatly encouraged us, and we appreciate your time and effort in reviewing our work.

---

### Official Review · Reviewer_UjLH · 2024-11-03

**Soundness:** 3
**Presentation:** 3
**Contribution:** 3
**Rating:** 8
**Confidence:** 4

**Summary:**

The authors presented an analysis of the contrastive loss of CLIP that gave insight on the optimal similarity measure for this loss. Furthermore, inspired by their analysis of the optimal loss, the authors proposed a new similarity measure that achives optimal similarity values, with reasonable computational cost.

**Strengths:**

The paper presents an alternative to the inner-product similarity through the use of weighted sums of kernel function evaluations. As such, the method benefits from the kernel trick: whereas the simple inner-product similarity may require a high-dimensionality in feature space, the use of kernels reduces the dimensionality of the feature space and shortens the amount of calculations required. Conversely, the method allows for greater performance in downstream tasks for the same amount of computation, by approsimating more closely the (theoretical) optimal similarity measure.

**Weaknesses:**

1. Inner-product similarity analysis

     A key point in the argument for the superiority of the authors' method over the standard similarity measure of CLIP is given in section 5.1, regarding the approximation error between the inner-product similarity matrix and the optimal similarity matrix $G$. There is an assumption that the number of samples $N$ is greater than the dimensionality $d$ of the feature space, thus the rank of the inner-product similarity matrix is smaller than the rank of $G$, implying the existence of approximation error. However, it is not clear from the paper that the approximation error is indeed deleterious, since the effective rank of $G$ could be substantially smaller than $N$ - and the effectiveness of CLIP suggests that this may indeed be the case.
     Another evidence in this direction is the fact that the authors had to use a combination of a linear kernel with a nonlinear kernel in their approach, and their ablation study showed that the linear-only kernel has comparable performance (by exceeding in the zero-shot classification task, and underperforming in the linear classification task, and in both cases by a small margin that is not evaluated in regards to its statistical significance).

2. Need for linear-only and nonlinear-only results

     The hyperparameter options for $(\alpha_1, \alpha_2)$ do not include a linear only $(1, 0)$ and nonlinear-only $(0,1)$ option. Given the ablation results, such choices were desirable.

**Questions:**

It would be interesting to have an analysis of:

- The effective rank of $G$ to clarify whether the stated deficiency of inner-product similarity is substantial. For instance, an analysis of the singular values of $G$ could be of value, since it would directly relate the rank disparity to the L2 error.

- The effectiveness of linear-only kernels in addition to the experiments already in place.

---

> ### Author Response · Authors · 2024-11-24
> **Response to Reviewer UjLH**
>
> We truly appreciate your constructive and insightful feedback. We hope that the following responses adequately address your concerns.
>
> > The effective rank of $G$ to clarify whether the stated deficiency of inner-product similarity is substantial. For instance, an analysis of the singular values of could be of value, since it would directly relate the rank disparity to the L2 error. (Weakness1: Inner-product similarity analysis)
>
> Thank you very much for your suggestion regarding an interesting research direction.
> Unfortunately, we were unable to conduct an analysis of the effective rank of $G$ in a practical case, since it is difficult to obtain the probability function of actual data. Additionally, obtaining a direct estimation of the probability density from generative models of pairs of texts and images still remains challenging. However, we agree that a practical analysis of real data is important and would be valuable for future work.
> Nonetheless, we can theoretically design an ideal similarity matrix $G$. For instance, we can list $N$ concepts (such as "dog", "cat", "apple", "desk", and so on) and assign a value of 1 to the similarity of text-image pairs that correspond to the same concept in the list, while assigning a value of 0 to the similarity of text-image pairs from different concepts. Consequently, the ideal similarity matrix will be the $N \times N$ identity matrix, which has a rank of $N$. When we consider combinations of concepts, such as "an apple on a desk" and "a dog under a desk," the number of concepts in the list could increase drastically. From this thought experiment, we believe that enhancing the representational capability of similarity is beneficial.
>
> > The effectiveness of linear-only kernels in addition to the experiments already in place. (Weakness 2: Need for linear-only and nonlinear-only results)
>
> * Regarding nonlinear-only $(0,1)$ cases, we found that the training of nonlinear-only models often became unstable, as we described around Lines 375-377. We have modified the manuscript to clarify this instability issue in Section 6.4. Additionally, we added the result of an ablation study using CC12M in Appendix A.4. On CC12M, training of nonlinear-only models, which we denote as WPCE Nonlinear, did not encounter a NaN loss issue, and we were able to present the performance of WPCE Nonlinear. We would appreciate it if you could take a look at this.
> * Regarding the comparable performance of the linear-only kernel, we apologize that the hyperparameter search was possibly insufficient. We additionally trained models with $(\alpha_1, \alpha_2) = (0.667, 0.333)$ and updated tables. As a result, WPCE Gaussian trained on CC3M scored 69.01% in the linear classification tasks, which exceeds by a margin of 1.61%. However, we acknowledge that WPCE Linear trained on CC3M still exceeds in the zero-shot classification task. While we currently think one possible reason is suboptimality of the prompts used in prompt ensembling for zero-shot classifications, we recognize that further analysis of this phenomena is important and could be future work.

---

> > ### Comment · Reviewer_UjLH · 2024-11-28
> >
> > On weakness 1: while I do agree that the rank of G is N in general, the effective rank [1] can be much smaller, that is, the error in reducing the dimensionality of the column space of G can actually be small. Take the given thought experiment provided in the response to comments:
> >
> > - If the similarity between paired vs non-paired items is indeed binary, then matrix G is the identity matrix, with rank N. But in the probabilistic similarity this is often not the case. Consider the concepts of "dog" and "cat", as given in the example. There is a lot of overlap between these concepts; therefore for many other concepts K, we have P(K | "dog") will not be too different from P(K | "cat"), and while the associated rows (or columns) of G may be different, they will not be radically different. Thus, for these two illustrating concepts, we can think of G as the sum of a matrix G' where the rows of "dog" and "cat" are identical (valued at the average of the original rows), plus a dG matrix encoding the difference. This procedure reduced the rank of G from N to the rank of G' which is (N-1) now, with the introduction of the small error dG.
> >
> > On weakness 2: I looked at the new results. It is difficult to know how significant the differences are, and the lack of convergence in the training procedure in some cases is also another point to investigate in future work - maybe the presence of the linear kernel has some regularizing effect on the training of the mixed linear/non-linear model? Anyway, I'm satisfied with the authors reply for this paper.
> >
> > All in all, I maintain my original assessment of the paper - it is interesting and well written, but there are some issues to investigate further, in a separate paper.
> >
> > [1] O. Roy and M. Vetterli, "The effective rank: A measure of effective dimensionality," 2007 15th European Signal Processing Conference, Poznan, Poland, 2007, pp. 606-610.

---

> > > ### Author Response · Authors · 2024-11-28
> > >
> > > Thank you for your valuable comments suggesting interesting directions for future research. We agree that there are still points to investigate in future work. In particular, analyses of the properties of similarity between real-world concepts induced by their probabilities, such as the effective rank, would be exciting both theoretically and practically.
> > >
> > > Once again, we sincerely appreciate your constructive and insightful feedback.

---

### Official Review · Reviewer_Sno6 · 2024-11-04

**Soundness:** 3
**Presentation:** 2
**Contribution:** 2
**Rating:** 6
**Confidence:** 4

**Summary:**

The paper presents a new understanding of the symmetric contrastive loss InfoNCE of CLIP, and clarify the point-wise mutual information is the optimal similarity that minimizes symmetric InfoNCE., then gives an upper bound of excess risk on downstream classification tasks of representations when the optimal similarity is achieved. Then, the paper proposes similarity based on weighted point clouds to consistently achieves the point-wise mutual information and the corresponding implementation of the proposed similarity.

**Strengths:**

1. The paper starts from a new understanding of contrastive loss, and proposes similarity measure that approximates the optimal formulation, which is theoretically well-founded.

2. The paper combines theory with practice and has demonstrated the effectiveness of the proposed similarity through experiments.

**Weaknesses:**

1. The innovation is limited. Some core concepts of the paper, such as suboptimality decomposition, are referenced from existing work.

2. The role of weighted point cloud seems to be overstated. The paper did not use concepts related to point clouds, such as neighborhoods, connectivity. Only the “weight” is used in the weighted similarity aggregation.

3. Some concepts or expressions need further clarification. For example, 1. What exactly does the similarity structure (line 41) mean? 2. Why is the rank of G greater than d+1, there exists a certain error of the approximation of G? (lines 309-310) 3. The assumption C.2 seems strong, can you provide a specific explanation based on practical examples? 4. The equation 3 has empty before ||, which seems a typo.

**Questions:**

See weakness.

---

> ### Author Response · Authors · 2024-11-24
> **Response to Reviewer Sno6 Part 1**
>
> We are grateful for your constructive reviews and insightful questions. We hope that our following answers adequately address your concerns.
>
> > The innovation is limited. Some core concepts of the paper, such as suboptimality decomposition, are referenced from existing work.
>
> We recognize that our theoretical results are derived using common techniques of error analysis, such as the risk decomposition in Eq. (4) and non-negativity of KL divergence.
> However, we would like to claim that our upper bound of excess risk is not trivial in that this is characterized by the capability of similarity class and the optimal similarity for the symmetric InfoNCE loss.
>
> > What exactly does the similarity structure (line 41) mean?
>
> We acknowledge that there exists a structural relationship between concepts determined by an ideal similarity matrix, which we have referred to as the "similarity structure."
> As emphasized in our paper, when utilizing Symmetric InfoNCE, the pointwise mutual information (PMI) serves as the ideal similarity matrix. However, we recognize that other loss functions may yield different ideal similarity matrices. Furthermore, it is indeed possible to conceive of various advantageous similarity matrices, for example, similarity matrices that explicitly take into account a knowledge graph of concepts. We anticipate that methods to realize these matrices will be explored in future research.
>
> > Why is the rank of G greater than d+1, there exists a certain error of the approximation of G? (lines 309-310)
>
> * If there is no error in the approximation of G, it holds that $G = Z_x^\top Z_y + \Gamma J $. However, it contradicts Proposition C.1, which ensures that the rank of $Z_x^\top Z_y + \Gamma J$ is less than $d+1$, and thus not equal to the rank of $G$ ($=N > d+1$, from the assumption in Line302). Therefore there exists a certain error of the approximation of G.
> * Here, we would like to provide a supplement to this discussion. Theorem 1 of [Georgieva and Hofreither, 2017] ensures that there exists a non-zero lower bound of the uniform error of low-rank matrix approximation. Therefore, unlike the case of our proposed similarity (our Theorem 5.1), the approximation error cannot be reduced arbitrarily.
>   - From Theorem 1 of [Georgieva and Hofreither, 2017], it holds that
>     $E_{\infty,\infty}^{N-1}(G) = \min_{h \neq 0}  \lVert G h \rVert_{\infty} / \lVert h \rVert_{1} $,
>     where $E^k_{\infty,\infty}(G) := \inf_{\mathrm{rank} A \leq k} \max_{i,j} |A_{ij} - G_{ij}|$ and $h \in \mathbb{R}^N$.
>   - Since it holds that $\lVert Gh \rVert_{\infty} / \lVert h \rVert_1 = \lVert G (h / \lVert h \rVert_1)\rVert_{\infty}$, we obtain $E_{\infty,\infty}^{N-1}(G) = \min_{\lVert h \rVert_1 = 1} \lVert G h \rVert_{\infty}$. Since $\mathrm{rank}~{G} = N$, we have $E_{\infty,\infty}^{N-1} (G) > 0$.
> $\lVert G h \rVert_{\infty}$ is continuous. $ \lVert h \rVert_1 = 1$ is closed. There exists a minimum, and it is not zero.
>   - For $k \leq N-1$, it holds that $E_{\infty,\infty}^k(G) \geq E_{\infty, \infty}^{N-1}(G)$.
>
> [Georgieva and Hofreither, 2017] I. Georgieva and C. Hofreither. "On best uniform approximation by low-rank matrices." Linear Algebra and its Applications 518 (2017): 159-176.
>
> > The role of weighted point cloud seems to be overstated. The paper did not use concepts related to point clouds, such as neighborhoods, connectivity.
>
> Here, we would like to clarify that we refer to a “weighted point cloud” as a set of pairs of a scalar weight and a vector point, and we have shown that the proposed similarity, under our definition of weighted point clouds, has a theoretical benefit in approximation capability of similarity matrices, which leads to advantage in downstream classification tasks.
> However, we also acknowledge that there is room for further investigation of more effective similarity metrics in practice. Incorporating concepts such as neighborhoods and connectivity into point-cloud-based similarities would be an interesting direction for future research. We are truly grateful for this insightful and interesting review.
>
> > The equation 3 has empty before ||, which seems a typo.
>
> We appreciate your pointing out the typo. We revised our draft accordingly.

---

> > ### Author Response · Authors · 2024-11-29
> >
> > I hope this message finds you well. Thank you again for reviewing our work. We would like to kindly check if you have any further concerns or if you found our responses helpful. We appreciate your time and feedback, and we look forward to hearing from you.

---

> > > ### Comment · Reviewer_Sno6 · 2024-11-29
> > >
> > > I have read the reply, and author's responses address my concerns.

---

> > > > ### Author Response · Authors · 2024-11-29
> > > >
> > > > Thank you for your response. We are glad that my responses addressed your concerns, and we appreciate your valuable feedback and engagement.

---

> ### Author Response · Authors · 2024-11-24
> **Response to Reviewer Sno6 Part 2**
>
> > The assumption C.2 seems strong, can you provide a specific explanation based on practical examples?
>
> * (a) assumes the existence of low-dimensional latent variables that data samples $X$ and $Y$ are generated from. We claim that this assumption is moderate and common in machine learning. For example, VAEs and GANs generate data from latent variables, which usually have less dimensionality than data domain.
> $Z^{(\mathcal{X})}$ and $Z^{(\mathcal{Y})}$ are mutually independent components of such latent variables. In an extreme case, $\tilde{X}$ and $\tilde{Y}$ can be $X$ and $Y$ themselves, respectively, and $d_\mathcal{X} = d_\mathcal{Y}=0$.
> * (b) assumes that there exist continuous bijective "decoders" that transform latent variables to data. Considering GANs and VAEs, whose decoders are typically modeled using continuous deep neural networks, we claim that the continuity of decoders is not a strong assumption.
> Regarding bijective property, we can achieve bijective decoders by appropriately restricting the support of latent variables. (Here, we assume that all data points can be generated by the decoder prior to this modification.) More precisely, if there are multiple latent variables that produce the same data, we select one of them, remove the others from the support of latent distribution, and renormalize the probability distribution. Through this process, we can obtain bijective decoders.
> * (c) assumes that the support of the data distribution of latent variables is compact. Since we consider Euclidean space, the supports of data distributions are compact if the supports are bounded and closed. (Note that supports of distributions are closed by definition.) Here, we consider cases of images and texts as an example and explain that the support of those latent distributions can be considered to be bounded.
>   - Each pixel of images is typically represented as a 3-dimensional real vector such that each entry is in the range of [0, 1]. Thus, the domain of images can be considered to be bounded. Therefore, it is reasonable to assume that the support of a latent distribution of images can be also bounded.
>   - Text can be represented as a sequence of words or tokens in a dictionary, and this dictionary typically has finite vocabulary. For simplicity, we consider texts of finite length, as typical generative models and representation models for texts are trained on text data with finite length. In this case, the number of possible texts is also finite. A set of finite discrete points is bounded, so it is reasonable to assume that the support of a latent distribution of texts is also bounded.
> * (d) assumes sufficient smoothness of the PMI for an analysis of approximation error. We acknowledge that this might be the strongest assumption in our paper, and relaxing this assumption would be a direction for future work. However, we would like to emphasize that this assumption encompasses sufficiently broad situations.
>   - To prevent PMI from being negative infinity, we assume that $p_{\tilde{X}, \tilde{Y}}(\tilde{x}, \tilde{y}) \neq 0$ for all $\tilde{x}, \tilde{y}$ such that $p_{\tilde{X}}(\tilde{x})p_{\tilde{Y}}(\tilde{y}) \neq 0$. While this could be somewhat strong theoretically, it can be considered to hold in practice. By introducing a sufficiently small $\epsilon > 0$ that is negligible in real-world observations and admitting $p_{\tilde{X},\tilde{Y}}(\tilde{X}, \tilde{Y}) > \epsilon$, we can avoid this issue.
>   - If $p_{\tilde{X}, \tilde{Y}}(\tilde{x}, \tilde{y})$ is Lipschitz continuous and not zero on the support of $p_{\tilde{X}} \times p_{\tilde{Y}}$, then PMI is also Lipschitz continuous. For example, distributions that are typically used as a prior distribution of VAEs and GANs, such as Gaussian distribution and a mixture of Gaussian distributions, are Lipschitz continuous.
>   - For simplicity, we have considered the case where $\tilde{X}$ and $\tilde{Y}$ are continuous and have a density probability function. However, even when $\tilde{X}$ and $\tilde{Y}$ are discrete, the same discussion holds, by using the Radon-Nikodym derivative as the definition of PMI and employing measures other than the Lebesgue measure, such as the Dirac measure, for the probability measure over the latent space.

---

### Author Response · Authors · 2024-11-24
**Message to the Reviewers**

Dear all reviewers,

We sincerely appreciate your taking the time to review our paper and are grateful for all the constructive and insightful feedback provided by the reviewers. In response to the feedback, we have made the following updates to our manuscript. The changes are highlighted in red in the updated version. We would appreciate it if you could take a look at it.
* We have corrected the typo in Theorem 4.2, as noted by Reviewers Sno6 and 2vyS.
* We have added explanations and results about ablation study, particularly regarding models that use only a non-linear kernel, as suggested by Reviewer UjLH. In conjunction with this additional experiment, we have expanded the hyperparameter search space for the coefficients $(\alpha_1, \alpha_2)$ and updated the corresponding tables.

---

### Meta-Review · Area_Chair_EWpJ · 2024-12-19

**Metareview:**

This work studies the pretraining and inference mechanisms of the alignment methods used in vision-language models (VLMs) such as CLIP. The authors provide new theoretical insights into the optimal form of embeddings to enhance downstream classification tasks and develop a kernel formulation based on these insights. Empirical evaluations demonstrate significant improvements over the vanilla form used in CLIP (based on cosine similarity). All reviewers acknowledge the novelty and significance of this study. The AC concurs with the reviewers' assessments and recommends the acceptance of this work, **congratulations**.

Please address the following comments in the final version:

- Explicitly explain the mechanics behind "[PAD] token being dependent on its relative position to [EOS]."
- Revise Equation 3, as the use of the dot symbol (e.g., $.\mathcal{Y}_c$) is confusing.
- Provide the parameters of the Random Fourier Features (RFF), specifically $\omega_t$ and $\beta_t$.
- Explain why the dimensionality of the RFF differs between pretraining and zero-shot evaluations.
- In computer vision, point cloud refers to inferring 3D shape of an object or scene from raw 3D points. The usage of a point cloud in this context to refer to a set of fixed-sized to represent data might not be optimal.
- The title of the paper may be revised. If insisting on the use of point cloud, a `:` should be placed after learning (Weighted Point Cloud Embedding for Multimodal Contrastive Learning: Toward an Optimal Similarity Metric)

**Additional Comments On Reviewer Discussion:**

Reviewers raised several concerns about the clarity of certain parts of the work. There were also inquiries regarding the performance of the method in scenarios not considered in the initial submission. The authors have addressed these concerns, and all reviewers agree that their questions have been satisfactorily resolved. Given the novelty of the work and the significance of the results, the AC recommends accepting the paper.

---

### Decision · Program_Chairs · 2025-01-22

Accept (Spotlight)